# Meta Optimal Transport

## Abstract

We study the use of amortized optimization to predict optimal transport (OT) maps
from the input measures, which we call *Meta OT*. This helps repeatedly solve sim-
ilar OT problems between different measures by leveraging the knowledge and in-
formation present from past problems to rapidly predict and solve new problems.
Otherwise, standard methods ignore the knowledge of the past solutions and sub-
optimally re-solve each problem from scratch. We instantiate Meta OT models in
discrete and continuous (Wasserstein-2) settings between images, spherical data,
and color palettes and use them to improve the computational time of standard OT
solvers by multiple orders of magnitude.

## 1 Introduction

Optimal transportation [Villani, 2009, Ambrosio, 2003, Santambrogio, 2015, Peyré et al., 2019,
Merigot and Thibert, 2021] is thriving in domains including economics [Galichon, 2016], rein-
forcement learning [Dadashi et al., 2021, Fickinger et al., 2021], style transfer [Kolkin et al., 2019],
generative modeling [Arjovsky et al., 2017, Seguy et al., 2018, Huang et al., 2020, Rout et al., 2021],
geometry [Solomon et al., 2015, Cohen et al., 2021], domain adaptation [Courty et al., 2017, Redko
et al., 2019], signal processing [Kolouri et al., 2017], fairness [Jiang et al., 2020], and cell repro-
gramming [Schiebinger et al., 2019]. A core component in these settings is to couple two measures
$(\alpha, \beta)$ supported on domains $(\mathcal{X}, \mathcal{Y})$ by solving a transport optimization problem such as the *primal
Kantorovich problem*, which is defined by:

$$\pi^\star(\alpha, \beta, c) \in \underset{\pi \in \mathcal{U}(\alpha, \beta)}{\arg\min} \int_{\mathcal{X} \times \mathcal{Y}} c(x, y) \mathrm{d}\pi(x, y), \tag{1}$$

where the *optimal coupling* $\pi^\star$ is a joint distribution over the product space, $\mathcal{U}(\alpha, \beta)$ is the set of
admissible couplings between $\alpha$ and $\beta$, and $c : \mathcal{X} \times \mathcal{Y} \to \mathbb{R}$ is the *ground cost*, that represents a
notion of distance between elements in $\mathcal{X}$ and elements in $\mathcal{Y}$.

**Challenges.** Unfortunately, solving eq. (1) *once* is computationally expensive between general mea-
sures and computationally cheaper alternatives are an active research topic: *Entropic optimal trans-
port* [Cuturi, 2013] smooths the transport problem with an entropy penalty, and *sliced distances*
[Kolouri et al., 2016, 2018, 2019, Deshpande et al., 2019] solve OT between 1-dimensional projec-
tions of the measures, where eq. (1) can be solved easily.

Furthermore, when an optimal transport method is deployed in practice, eq. (1) is not just solved
a single time, but is *repeatedly* solved for new scenarios between different input measures $(\alpha, \beta)$.
For example, the measures could be representations of images we care about optimally transporting
between and in deployment we would receive a stream of new images to couple. Repeatedly solving
optimal transport problems also comes up in the context of comparing seismic signals [Engquist
and Froese, 2013] and in single-cell perturbations [Bunne et al., 2021, 2022b,a]. Standard optimal
transport solvers deployed in this setting would re-solve the optimization problems from scratch, but
this ignores the shared structure and information present between different coupling problems.

**Overview and outline.** We study the use of amortized optimization and machine learning methods to rapidly solve multiple optimal transport problems and predict the solution from the input measures $(\alpha, \beta)$. This setting involves learning a *meta* model to predict the solution to the optimal transport problem, which we will refer to as *Meta Optimal Transport*. We learn Meta OT models to predict the solutions to optimal transport problems and significantly improve the computational time and number of iterations needed to solve eq. (1) between discrete (sect. 3.1) and continuous (sect. 3.2) measures. The paper is organized as follows: sect. 2 recalls the main concepts needed for the rest of the paper, in particular the formulations of the entropy regularized and unregularized optimal transport problems and the basic notions of amortized optimization; sect. 3 presents the Meta OT models and algorithms; and sect. 4 empirically demonstrates the effectiveness of Meta OT.

**Settings that are not Meta OT.** Meta OT is not useful in OT settings that do *not* involve *repeatedly* solving OT problems over a fixed distribution, including 1) standard generative modeling settings, such as Arjovsky et al. [2017] that estimate the OT distance between the data and model distributions, and 2) the out-of-sample setting of Seguy et al. [2018], Perrot et al. [2016] that couple measures and then extrapolate the map to larger measures containing the original measures.

## 2 Preliminaries and background

### 2.1 Dual optimal transport solvers

We review foundations of optimal transportation, following the notation of Peyré et al. [2019] in most places. The discrete setting often favors the entropic regularized version since it can be computed efficiently and in a parallelized way using the Sinkhorn algorithm. On the other hand, the continuous setting is often solved from samples using convex potentials. While the primal Kantorovich formulation in eq. (1) provides an intuitive problem description, optimal transport problems are rarely solved directly in this form due to the high-dimensionality of the couplings $\pi$ and the difficulty of satisfying the coupling constraints $\mathcal{U}(\alpha, \beta)$. Instead, most computational OT solvers use the *dual* of eq. (1), which we build our Meta OT solvers on top of in discrete and continuous settings.

#### 2.1.1 Entropic OT between discrete measures with the Sinkhorn algorithm

Let $\alpha := \sum_{i=1}^m a_i \delta_{x_i}$ and $\beta := \sum_{i=1}^n b_i \delta_{y_i}$ be *discrete* measures, where $\delta_z$ is a Dirac at point $z$ and $a \in \Delta_{m-1}$ and $b \in \Delta_{n-1}$ are in the *probability simplex* defined by

$$\Delta_{k-1} := \{x \in \mathbb{R}^k : x \geq 0 \text{ and } \sum_i x_i = 1\}. \quad (2)$$

---

**Algorithm 1** Sinkhorn($\alpha, \beta, c, \epsilon, f_0 = 0$)

---

**for** iteration $i = 1$ to $N$ **do**
    $g_i \leftarrow \epsilon \log b - \epsilon \log \left(K^\top \exp\{f_{i-1}/\epsilon\}\right)$
    $f_i \leftarrow \epsilon \log a - \epsilon \log \left(K \exp\{g_i/\epsilon\}\right)$
**end for**
Compute $P_N$ from $f_N, g_N$ using eq. (6)
**return** $P_N \approx P^\star$

---

**Discrete OT.** In the discrete setting, eq. (1) simplifies to the *linear program*

$$P^\star(\alpha, \beta, c) \in \operatorname*{arg\,min}_{P \in U(a,b)} \langle C, P \rangle \qquad U(a,b) := \{P \in \mathbb{R}_+^{n \times m} : P1_m = a, \quad P^\top 1_n = b\} \quad (3)$$

where $P$ is a *coupling matrix*, $P^\star(\alpha, \beta)$ is the *optimal* coupling, and the *cost* can be discretized as a matrix $C \in \mathbb{R}^{m \times n}$ with entries $C_{i,j} := c(x_i, y_j)$, and $\langle C, P \rangle := \sum_{i,j} C_{i,j} P_{i,j}$,

**Entropic OT.** The linear program above can be regularized adding the entropy of the coupling to smooth the objective as in Cominetti and Martín [1994], Cuturi [2013], resulting in:

$$P^\star(\alpha, \beta, c, \epsilon) \in \operatorname*{arg\,min}_{P \in U(a,b)} \langle C, P \rangle - \epsilon H(P) \quad (4)$$

where $H(P) := -\sum_{i,j} P_{i,j}(\log(P_{i,j}) - 1)$ is the discrete entropy of a coupling matrix $P$.

**Entropic OT dual.** As presented in Peyré et al. [2019, Prop. 4.4], the dual of eq. (4) is

$$f^\star, g^\star \in \operatorname*{arg\,max}_{f \in \mathbb{R}^n, g \in \mathbb{R}^m} \langle f, a \rangle + \langle g, b \rangle - \epsilon \langle \exp\{f/\epsilon\}, K \exp\{g/\epsilon\} \rangle, \quad K_{i,j} := \exp\{-C_{i,j}/\epsilon\}, \quad (5)$$

where $K \in \mathbb{R}^{m \times n}$ is the *Gibbs kernel* and the *dual variables* or *potentials* $f \in \mathbb{R}^n$ and $g \in \mathbb{R}^m$ are associated, respectively, with the marginal constraints $P1_m = a$ and $P^\top 1_n = b$. The optimal duals depend on the problem, e.g. $f^\star(\alpha, \beta, c, \epsilon)$, but we omit this dependence for notational simplicity.

**Recovering the primal solution from the duals.** Given optimal duals $f^\star$, $g^\star$ that solve eq. (5) the optimal coupling $P^\star$ to the primal problem in eq. (4) can be obtained by

$$P_{i,j}^\star(\alpha,\beta,c,\epsilon) := \exp\{f_i^\star/\epsilon\}K_{i,j}\exp\{g_j^\star/\epsilon\} \qquad (K \text{ is defined in eq. (5)}) \tag{6}$$

**The Sinkhorn algorithm.** Algorithm 1 summarizes the log-space version, which takes closed-form block coordinate ascent updates on eq. (5) obtained from the first-order optimality conditions [Peyré et al., 2019, Remark 4.21]. We will use it to fine-tune predictions made by our Meta OT models.

**Computing the error.** Standard implementations of the Sinkhorn algorithm, such as Flamary et al. [2021], Cuturi et al. [2022], measure the error of a candidate dual solution $(f,g)$ by computing the deviation from the marginal constraints, which we will also use in comparing our solution quality:

$$\text{err}(f,g;\alpha,\beta,c) := \|P\mathbf{1}_m - a\|_1 + \|P^\top\mathbf{1}_n - b\|_1 \qquad (\text{compute } P \text{ from eq. (6)}) \tag{7}$$

**Mapping between the duals.** The first-order optimality conditions of eq. (5) also provide an equivalence between the optimal dual potentials that we will make use of:

$$g(f;b,c) := \epsilon\log b - \epsilon\log\left(K^\top\exp\{f/\epsilon\}\right). \tag{8}$$

### 2.1.2 Wasserstein-2 OT between continuous (Euclidean) measures with dual potentials

Let $\alpha$ and $\beta$ be continuous measures in Euclidean space $\mathcal{X} = \mathcal{Y} = \mathbb{R}^d$ (with $\alpha$ absolutely continuous with respect to the Lebesgue measure) and the ground cost be the squared Euclidean distance $c(x,y) := \|x-y\|_2^2$. Then the minimum of eq. (1) defines the square of the *Wasserstein-2* distance:

---
**Algorithm 2** W2GN$(\alpha,\beta,\varphi_0)$

---
**for** iteration $i = 1$ to $N$ **do**
    Sample from $(\alpha,\beta)$ and estimate $\mathcal{L}(\varphi_{i-1})$
    Update $\varphi_i$ with approximation to $\nabla_\varphi\mathcal{L}(\varphi_{i-1})$
**end for**
**return** $T_N(\cdot) := \nabla_x\psi_{\varphi_N}(\cdot) \approx T^\star(\cdot)$

---

$$W_2^2(\alpha,\beta) := \min_{\pi\in\mathcal{U}(\alpha,\beta)}\int_{\mathcal{X}\times\mathcal{Y}}\|x-y\|_2^2\mathrm{d}\pi(x,y) = \min_T\int_{\mathcal{X}}\|x-T(x)\|_2^2\mathrm{d}\alpha(x), \tag{9}$$

where $T$ is a *transport map* pushing $\alpha$ to $\beta$, i.e. $T_\#\alpha = \beta$ with the *pushforward operator* defined by $T_\#\alpha(B) := \alpha(T^{-1}(B))$ for any measurable set $B$.

**Convex dual potentials.** The primal form in eq. (9) is difficult to solve, as in the discrete setting, due to the difficulty of representing the coupling and satisfying the constraints. Makkuva et al. [2020], Taghvaei and Jalali [2019], Korotin et al. [2019, 2021b, 2022] propose to instead solve the dual:

$$\psi^\star(\,\cdot\,;\alpha,\beta) \in \underset{\psi\in\text{convex}}{\arg\min}\int_{\mathcal{X}}\psi(x)\mathrm{d}\alpha(x) + \int_{\mathcal{Y}}\overline{\psi}(y)\mathrm{d}\beta(y), \tag{10}$$

where $\psi$ is a convex function referred to as a *convex potential*, and $\overline{\psi}(y) := \max_{x\in\mathcal{X}}\langle x,y\rangle - \psi(x)$ is the *Legendre-Fenchel transform* or *convex conjugate* of $\psi$ [Fenchel, 1949, Rockafellar, 2015]. The potential $\psi$ is often approximated with an input-convex neural network (ICNN) [Amos et al., 2017].

**Recovering the primal solution from the dual.** Given an optimal dual $\psi^\star$ for eq. (10), Brenier [1991] remarkably shows that an optimal map $T^\star$ for eq. (9) can be obtained with differentiation:

$$T^\star(x) = \nabla_x\psi^\star(x). \tag{11}$$

**Wasserstein-2 Generative Networks (W2GNs).** Korotin et al. [2019] model $\psi_\varphi$ and $\overline{\psi_\varphi}$ in eq. (10) with two separate ICNNs parameterized by $\varphi$. The separate model for $\overline{\psi_\varphi}$ is useful because the conjugate operation in eq. (10) becomes computationally expensive. They optimize the loss:

$$\mathcal{L}(\varphi) := \underbrace{\mathop{\mathbb{E}}_{x\sim\alpha}[\psi_\varphi(x)] + \mathop{\mathbb{E}}_{y\sim\beta}\left[\langle\nabla\overline{\psi_\varphi}(y),y\rangle - \psi_\varphi(\nabla\overline{\psi_\varphi}(y))\right]}_{\text{Cyclic monotone correlations (dual objective)}} + \gamma\underbrace{\mathop{\mathbb{E}}_{y\sim\beta}\|\nabla\psi_\varphi\circ\nabla\overline{\psi_\varphi}(y) - y\|_2^2}_{\text{Cycle-consistency regularizer}}, \tag{12}$$

where $\overline{\varphi}$ is a detached copy of the parameters and $\gamma$ is a hyper-parameter. The first term are the *cyclic monotone correlations* [Chartrand et al., 2009, Taghvaei and Jalali, 2019], that optimize the dual objective in eq. (10), and the second term provides *cycle consistency* [Zhu et al., 2017] to estimate the conjugate $\overline{\psi}$. Algorithm 2 shows how $\mathcal{L}$ is typically optimized using samples from the measures, which we use to fine-tune Meta OT predictions.

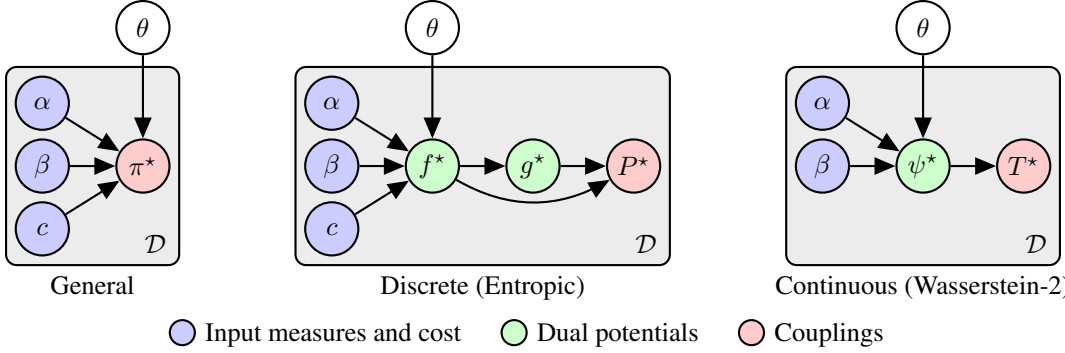

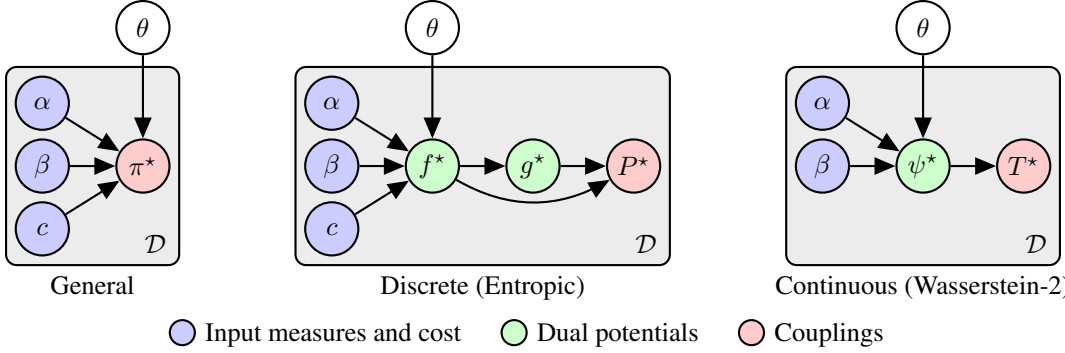

◯ Input measures and cost    ◯ Dual potentials    ◯ Couplings

Figure 1: Meta OT uses objective-based amortization for optimal transport. In the general formulation, the *parameters* $\theta$ capture shared structure in the *optimal couplings* $\pi^\star$ between multiple input measures and costs over some *distribution* $\mathcal{D}$. In practice, we learn this shared structure over the *dual potentials* which map back to the coupling: $f^\star$ in discrete settings and $\psi^\star$ in continuous ones.

## 2.2 Amortized optimization and learning to optimize

Our paper is an application of amortized optimization methods that predict the solutions of optimization problems, as surveyed in, e.g., Chen et al. [2021], Amos [2022]. We use the basic setup from Amos [2022], which considers unconstrained continuous optimization problems of the form

$$z^\star(\phi) \in \arg\min_z J(z; \phi), \tag{13}$$

where $J$ is the objective, $z \in \mathcal{Z}$ is the *domain*, and $\phi \in \Phi$ is some *context* or *parameterization*. In other words, the context conditions the objective but is not optimized over. Given a *distribution over contexts* $\mathcal{P}(\phi)$, we learn a model $\hat{z}_\theta$ parameterized by $\theta$ to approximate eq. (13), i.e. $\hat{z}_\theta(\phi) \approx z^\star(\phi)$. $J$ will be differentiable for us, so we optimize the parameters using *objective-based learning* with

$$\min_\theta \mathbb{E}_{\phi \sim \mathcal{P}(\phi)} J(\hat{z}_\theta(\phi); \phi), \tag{14}$$

which does *not* require ground-truth solutions $z^\star$ and can be optimized with a gradient-based solver. While we focus on optimizing eq. (14) because we do not assume easy access to ground-truth solutions $z^\star(\phi)$, one alternative is *regression-based learning* if the solutions are easily available:

$$\min_\theta \mathbb{E}_{\phi \sim \mathcal{P}(\phi)} \|z^\star(\phi) - \hat{z}_\theta(\phi)\|_2^2. \tag{15}$$

## 3 Meta Optimal Transport

Figure 1 illustrates our key contribution of connecting objective-based amortization in eq. (14) to optimal transport. We consider solving *multiple* OT problems and learning shared structure and correlations between them. We denote a joint *meta-distribution* over the input measures and costs with $\mathcal{D}(\alpha, \beta, c)$, which we call *meta* to distinguish it from the measures $\alpha, \beta$.

In general, we could introduce a model that directly predicts the primal solution to eq. (1), i.e. $\pi_\theta(\alpha, \beta, c) \approx \pi^\star(\alpha, \beta, c)$ for $(\alpha, \beta, c) \sim \mathcal{D}$. This is difficult for the same reason why most computational methods do not operate directly in the primal space: the optimal coupling is often a high-dimensional joint distribution with non-trivial marginal constraints. We instead turn to predicting the dual variables used by today's solvers.

### 3.1 Meta OT between discrete measures

We build on standard methods for entropic OT reviewed in sect. 2.1.1 between discrete measures $\alpha := \sum_{i=1}^m a_i \delta_{x_i}$ and $\beta := \sum_{i=1}^n b_i \delta_{x_i}$ with $a \in \Delta_{m-1}$ and $b \in \Delta_{n-1}$ coupled using a cost $c$. In the Meta OT setting, the measures and cost are the contexts for amortization and sampled from a *meta-distribution*, i.e. $(\alpha, \beta, c) \sim \mathcal{D}(\alpha, \beta, c)$. For example, sects. 4.1 and 4.2 considers meta-distributions over the weights of the atoms, i.e. $(a, b) \sim \mathcal{D}$, where $\mathcal{D}$ is a distribution over $\Delta_{m-1} \times \Delta_{n-1}$.

| **Algorithm 3** Training Meta OT | **Algorithm 4** Fine-tuning with Sinkhorn |
|---|---|

**Algorithm 3** Training Meta OT

Initialize amortization model with $\theta_0$
**for** iteration $i$ **do**
    Sample $(\alpha, \beta, c) \sim \mathcal{D}$
    Predict duals $\hat{f}_\theta$ or $\hat{\varphi}_\theta$ on the sample
    Estimate the loss in eq. (17) or eq. (18)
    Update $\theta_{i+1}$ with a gradient step
**end for**

**Algorithm 4** Fine-tuning with Sinkhorn

Predict duals $\hat{f}_\theta(\alpha, \beta, c)$
**return** Sinkhorn$(\alpha, \beta, c, \epsilon, \hat{f}_\theta)$

**Algorithm 5** Fine-tuning with W2GN

Predict dual ICNN parameters $\hat{\varphi}_\theta(\alpha, \beta, c)$
**return** W2GN$(\alpha, \beta, c, T, \hat{\varphi}_\theta)$

**Amortization objective.** We will seek to predict the *optimal* potential. At optimality, the pair of potentials are related to each other via eq. (8), i.e. $g(f; \alpha, \beta, c) := \epsilon \log b - \epsilon \log \left( K^\top \exp\{f/\epsilon\} \right)$ where $K \in \mathbb{R}^{m \times n}$ is the *Gibbs kernel* from eq. (5). Hence, it is sufficient to predict one of the potentials, e.g. $f$, and recover the other. We thus re-formulate eq. (5) to just optimize over $f$ with

$$f^\star(\alpha, \beta, c, \epsilon) \in \underset{f \in \mathbb{R}^n}{\arg\min} \; J(f; \alpha, \beta, c), \tag{16}$$

where $-J(f; \alpha, \beta, c) := \langle f, a \rangle + \langle g, b \rangle - \epsilon \langle \exp\{f/\epsilon\}, K \exp\{g/\epsilon\} \rangle$ is the (negated) dual objective. Even though most solvers optimize over $f$ and $g$ jointly as in eq. (16), amortizing over these would likely need: 1) to have a higher capacity than a model just predicting $f$, and 2) to learn how $f$ and $g$ are connected through eq. (8) while in eq. (16) we explicitly provide this knowledge.

**Amortization model.** We predict the solution to eq. (16) with $\hat{f}_\theta(\alpha, \beta, c)$ parameterized by $\theta$, resulting in a computationally efficient approximation $\hat{f}_\theta \approx f^\star$. Here we use the notation $\hat{f}_\theta(\alpha, \beta, c)$ to mean that the model $\hat{f}_\theta$ depends on *representations* of the input measures and cost. In our settings, we define $\hat{f}_\theta$ as a fully-connected MLP mapping from the atoms of the measures to the duals.

**Amortization loss.** Applying objective-based amortization from eq. (14) to the dual in eq. (16) completes our learning setup. Our model should best-optimize the expectation of the dual objective

$$\min_\theta \; \underset{(\alpha, \beta, c) \sim \mathcal{D}}{\mathbb{E}} J(\hat{f}_\theta(\alpha, \beta, c); \alpha, \beta, c), \tag{17}$$

which is appealing as it does not require ground-truth solutions $f^\star$. Algorithm 3 shows a basic training loop for eq. (17) using a gradient-based optimizer such as Adam [Kingma and Ba, 2014].

**Sinkhorn fine-tuning.** The dual prediction made by $\hat{f}_\theta$ with an associated $\hat{g}$ can easily be input as the initialization to a standard Sinkhorn solver as shown in algorithm 4. This allows us to deploy the predicted potential with Sinkhorn to obtain the optimal potentials with only a few extra iterations.

**On accelerated solvers.** Here we have only considered fine-tuning the Meta OT prediction with a log-Sinkhorn solver. Meta OT can also be combined with accelerated variants of entropic OT solvers such as Thibault et al. [2017], Altschuler et al. [2017], Alaya et al. [2019], Lin et al. [2019] that would otherwise solve every problem from scratch.

## 3.2 Meta OT between continuous measures (Wasserstein-2)

We take an analogous approach to predicting the Wasserstein-2 map between continuous measures for Wasserstein-2 as reviewed in sect. 2.1.2. Here the measures $\alpha, \beta$ are supported in continuous space $\mathcal{X} = \mathcal{Y} = \mathbb{R}^d$ and we focus on computing Wasserstein-2 couplings from instances sampled from a *meta-distribution* $(\alpha, \beta) \sim \mathcal{D}(\alpha, \beta)$. The cost $c$ is not included in $\mathcal{D}$ as it remains fixed to the squared Euclidean cost everywhere here.

One challenge here is that the optimal dual potential $\psi^\star(\,\cdot\,; \alpha, \beta)$ in eq. (10) is a convex function and not simply a finite-dimensional real vector. The dual potentials in this setting are approximated by, e.g., an ICNN. We thus propose a *Meta ICNN* that predicts the *parameters* $\varphi$ of an ICNN $\psi_\varphi$ that approximates the optimal dual potentials, which can be seen as a hypernetwork [Stanley et al., 2009, Ha et al., 2016]. The dual prediction made by $\hat{\varphi}_\theta$ can easily be input as the initial value to a standard W2GN solver as shown in algorithm 5. App. B discusses other modeling choices we considered: we tried models based on MAML [Finn et al., 2017] and neural processes [Garnelo et al., 2018b,a].

Sinkhorn (converged, ground-truth)  Meta OT (initial prediction)

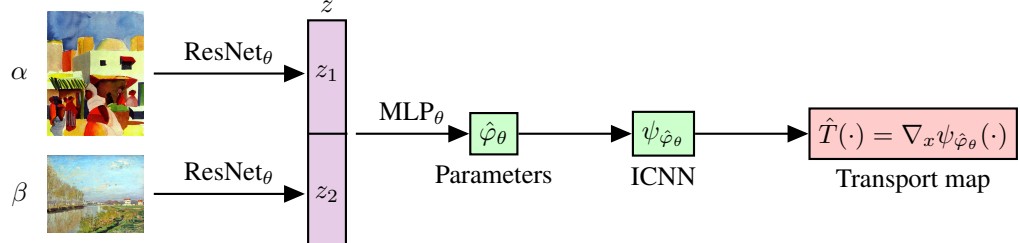

$\alpha_0 \longleftarrow \alpha_1 \longrightarrow \alpha_2$   $\alpha_0 \longleftarrow \alpha_1 \longrightarrow \alpha_2$

Figure 2: Interpolations between MNIST test digits using couplings obtained from (left) solving the problem with Sinkhorn, and (right) Meta OT model's initial prediction, which is $\approx$**100** times computationally cheaper and produces a nearly identical coupling.

Figure 3: A Meta ICNN for image-based input measures. A shared ResNet processes the input measures $\alpha$ and $\beta$ into latents $z$ that are decoded with an MLP into the parameters $\varphi$ of an ICNN dual potential $\psi_\varphi$. The derivative of the ICNN provides the transport map $\hat{T}$.

Table 1: Sinkhorn runtime (seconds) to reach a marginal error of $10^{-3}$. Meta OT's initial prediction takes $\approx 5 \cdot 10^{-5}$ seconds.

| Initialization | MNIST | Spherical |
|---|---|---|
| Zeros | $7.7 \cdot 10^{-3} \pm 1.2 \cdot 10^{-3}$ | $1.4 \pm 1.9 \cdot 10^{-1}$ |
| Gaussian | $7.7 \cdot 10^{-3} \pm 1.4 \cdot 10^{-3}$ | $1.1 \pm 2.0 \cdot 10^{-1}$ |
| Meta OT | $3.9 \cdot 10^{-3} \pm 1.6 \cdot 10^{-3}$ | $0.44 \pm 1.5 \cdot 10^{-1}$ |

Table 2: Color transfer runtimes and values.

| | Iter | Runtime (s) | Dual Value |
|---|---|---|---|
| Meta OT + W2GN | None | $3.5 \cdot 10^{-3} \pm 2.7 \cdot 10^{-4}$ | $0.90 \pm 6.08 \cdot 10^{-2}$ |
| | 1k | $0.93 \pm 2.27 \cdot 10^{-2}$ | $1.0 \pm 2.57 \cdot 10^{-3}$ |
| | 2k | $1.84 \pm 3.78 \cdot 10^{-2}$ | $1.0 \pm 5.30 \cdot 10^{-3}$ |
| W2GN | 1k | $0.90 \pm 1.62 \cdot 10^{-2}$ | $0.96 \pm 2.62 \cdot 10^{-2}$ |
| | 2k | $1.81 \pm 3.05 \cdot 10^{-2}$ | $0.99 \pm 1.14 \cdot 10^{-2}$ |

We report the mean and standard deviation across 10 test instances.

**Amortization objective.** We build on the W2GN formulation [Korotin et al., 2019] and seek parameters $\varphi^\star$ optimizing the dual ICNN potentials $\psi_\varphi$ and $\overline{\psi_\varphi}$ with $\mathcal{L}(\varphi; \alpha, \beta)$ from eq. (12). We chose W2GN due to the stability, but could also easily use other losses optimizing ICNN potentials.

**Amortization model: the Meta ICNN.** We predict the solution to eq. (12) with $\hat{\varphi}_\theta(\alpha, \beta)$ parameterized by $\theta$, resulting in a computationally efficient approximation to the optimum $\hat{\varphi}_\theta \approx \varphi^\star$. Figure 3 instantiates a convolutional Meta ICNN model using a ResNet-18 [He et al., 2016] architecture for coupling image-based measures. We again emphasize that $\alpha, \beta$ used with the model here are *representations* of measures, which in our cases are simply images.

**Amortization loss.** Applying objective-based amortization from eq. (14) to the W2GN loss in eq. (12) completes our learning setup. Our model should best-optimize the expectation of the loss:

$$\min_\theta \ \mathop{\mathbb{E}}_{(\alpha,\beta)\sim\mathcal{D}} \mathcal{L}(\hat{\varphi}_\theta(\alpha, \beta); \alpha, \beta). \tag{18}$$

As in the discrete setting, it does not require ground-truth solutions $\varphi^\star$ and we learn it with Adam.

## 4   Experiments

We demonstrate how Meta OT models improve the convergence of the state-of-the-art solvers in settings where solving multiple OT problems naturally arises. We implemented our code in JAX [Bradbury et al., 2018] as an extension to the the Optimal Transport Tools (OTT) package [Cuturi et al., 2022]. App. C covers further experimental and implementation details, and shows that all of our experiments take a few hours to run on our single Quadro GP100 GPU.

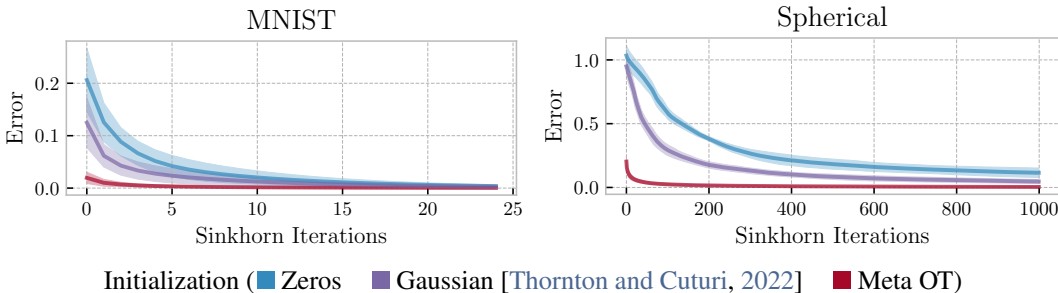

Figure 4: Meta OT successfully predicts warm-start initializations that significantly improve the convergence of Sinkhorn iterations on test data. The error is the marginal error defined in eq. (7).

## 4.1 Discrete OT between MNIST digits

Images provide a natural setting for Meta OT where the distribution over images provide the meta-distribution $\mathcal{D}$ over OT problems. Given a pair of images $\alpha_0$ and $\alpha_1$, each grayscale image is cast as a discrete measure in 2-dimensional space where the intensities define the probabilities of the atoms. The goal is to compute the optimal transport interpolation between the two measures as in, e.g., Peyré et al. [2019, §7]. Formally, this means computing the optimal coupling $P^\star$ by solving the entropic optimal transport problem between $\alpha_0$ and $\alpha_1$ and computing the interpolates as $\alpha_t = (t\operatorname{proj}_y + (1-t)\operatorname{proj}_x)_{\#}P^\star$, for $t \in [0,1]$, where $\operatorname{proj}_x(x,y) := x$ and $\operatorname{proj}_y(x,y) = y$. We selected $\epsilon = 10^{-2}$ as app. A shows that it gives interpolations that are not too blurry or sharp.

Our Meta OT model $\hat{f}_\theta$ (sect. 3.1) is an MLP that predicts the transport map between pairs of MNIST digits. We train on every pair from the standard training dataset. Figure 2 shows that even without fine-tuning, Meta OT's predicted Wasserstein interpolations between the measures are close to the ground-truth interpolations obtained from running the Sinkhorn algorithm to convergence. We then fine-tune Meta OT's prediction with Sinkhorn as in algorithm 4. Figure 4 shows that the near-optimal predictions can be quickly refined in fewer iterations than running Sinkhorn with the default initialization, and table 1 shows the runtime required to reach the default threshold, which uses the default marginal error threshold of $10^{-3}$. We compare our learned initialization to the standard zero initialization, as well as the Gaussian initialization proposed in Thornton and Cuturi [2022], which takes a continuous Gaussian approximation of the measures and initializes the potentials to be the known coupling between the Gaussians. This Gaussian initialization assumes the squared Euclidean cost, which is not the case in our spherical transport problem, but we find it is still helpful over the zero initialization.

## 4.2 Discrete OT for supply-demand transportation on spherical data

We next set up a synthetic transport problem between supply and demand locations where the supply and demands may change locations or quantities frequently, creating another Meta OT setting to be able to rapidly solve the new instances. We specifically consider measures living on the 2-sphere defined by $\mathcal{S}_2 := \{x \in \mathbb{R}^3 : \|x\| = 1\}$, i.e. $\mathcal{X} = \mathcal{Y} = \mathcal{S}_2$, with the transport cost given by the spherical distance $c(x,y) = \arccos(\langle x,y\rangle)$. We then randomly sample supply locations uniformly from Earth's landmass and demand locations from Earth's population density to induce a class of transport problems on the sphere obtained from the CC-licensed dataset from Doxsey-Whitfield et al. [2015]. Figure 5 shows that the predicted transport maps on test instances are close to the optimal maps obtained from Sinkhorn to convergence. Similar to the MNIST setting, fig. 4 and table 1 show improved convergence and runtime.

## 4.3 Continuous Wasserstein-2 color transfer

The problem of color transfer between two images consists in mapping the color palette of one image into the other one. The images are required to have the same number of channels, for example RGB images. The continuous formulation that we use from Korotin et al. [2019], takes i.e. $\mathcal{X} = \mathcal{Y} = [0,1]^3$ with $c$ being the squared Euclidean distance. We collected $\approx$200 public domain images from WikiArt and trained a Meta ICNN model from sect. 3.2 to predict the color transfer maps between

Sinkhorn (converged, ground-truth)  Meta OT (initial prediction)

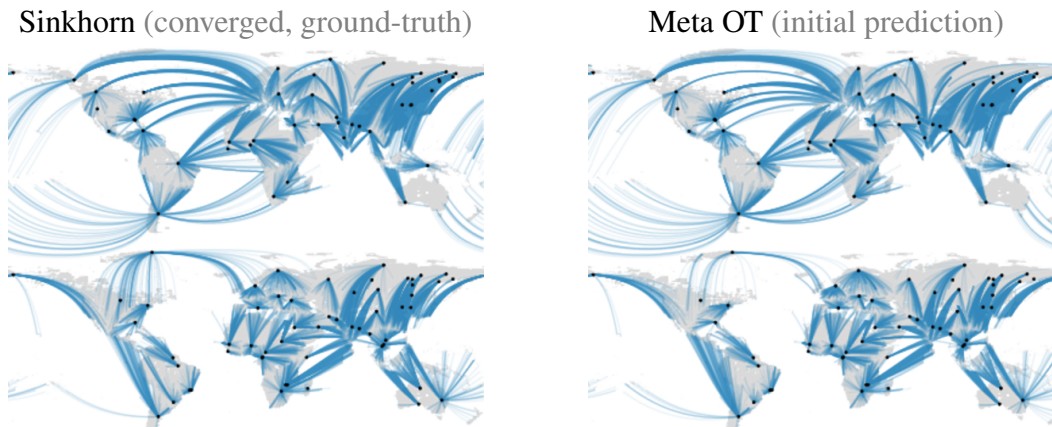

Figure 5: Test set coupling predictions of the spherical transport problem. Meta OT's initial prediction is ≈**37500** times faster than solving Sinkhorn to optimality. Supply locations are shown as black dots and the blue lines show the spherical transport maps $T$ going to demand locations at the end. The sphere is visualized with the Mercator projection.

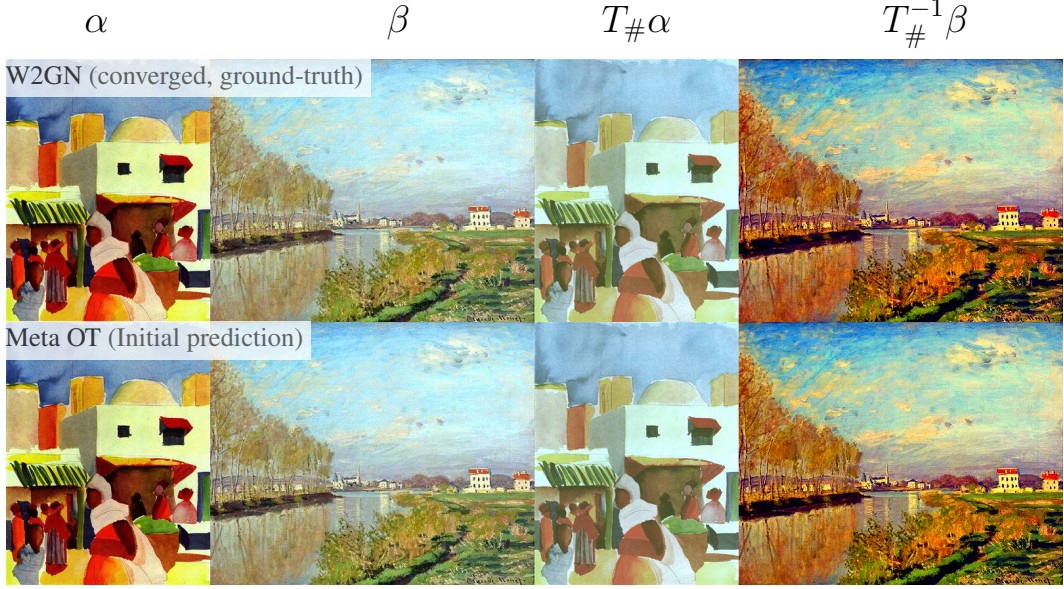

Figure 6: Color transfers with a Meta ICNN on test pairs of images. The objective is to optimally transport the continuous RGB measure of the first image $\alpha$ to the second $\beta$, producing an invertible transport map $T$. Meta OT's prediction is ≈**1000** times faster than training W2GN from scratch. The image generating $\alpha$ is Market in Algiers by August Macke (1914) and $\beta$ is Argenteuil, The Seine by Claude Monet (1872), obtained from WikiArt.

every pair of them. Figure 6 shows the predictions on test pairs and fig. 7 shows the convergence in comparison to the standard W2GN learning. Table 2 reports runtimes and app. E shows additional results.

## 5   Related work

**Efficiently estimating OT maps.** To compute OT maps with fixed cost between pairs of measures efficiently, neural OT models [Korotin et al., 2019, Li et al., 2020, Korotin et al., 2021a, Mokrov et al., 2021, Korotin et al., 2021b] leverage ICNNs to estimate maps between continuous high-

dimensional measures given samples from these, and Litvinenko et al. [2021], Scetbon et al. [2021], Forrow et al. [2019], Sommerfeld et al. [2019], Scetbon et al. [2022], Muzellec and Cuturi [2019], Bonet et al. [2021] leverage structural assumptions on coupling and cost matrices to reduce the computational and memory complexity. In the meta-OT setting, we consider learning to rapidly compute OT mappings between new pairs measures. All these works can hence potentially benefit from an acceleration effect by leveraging amortization similarly.

**Embedding measures where OT distances are discriminative.** Effort has been invested in learning encodings/projections of measures through a nested optimization problem, which aims to find discriminative embeddings of the measures to be compared [Genevay et al., 2018, Deshpande et al., 2019, Nguyen and Ho, 2022]. While these works share an encoder and/or a projection across task with the aim of leveraging more discriminative alignments (and hence an OT distance with a metric different from the Euclidean metric), our work differs in the sense that we find good initializations to solve the OT problem itself with fixed cost more efficiently across tasks.

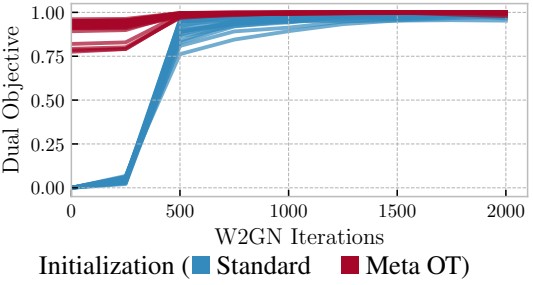

Figure 7: Convergence on color transfer test instances using W2GN. Meta ICNNs predicts warm-start initializations that significantly improve the (normalized) dual objective values.

**Optimal transport and amortization.** Few previous works in the OT literature leverage amortization. Courty et al. [2018] learn a latent space in which the Wasserstein distance between the measure's embeddings is equivalent to the Euclidean distance. Concurrent work [Nguyen and Ho, 2022] amortizes the estimation of the optimal projection in the max-sliced objective, which differs from our work where we instead amortize the estimation of the optimal coupling directly. Also, Lacombe et al. [2021] learns to predict Wasserstein barycenters of pixel images by training a convolutional networks that, given images as input, outputs their barycenters. Our work is hence a generalization of this pixel-based work to general measures – both discrete and continuous. A limitation of Lacombe et al. [2021] is that it does not provide alignments, as the amortization networks predicts the barycenter directly rather than individual couplings.

# 6 Conclusions, future directions, and limitations

We have presented foundations for modeling and learning to solve OT problems with Meta OT by using amortized optimization to predict optimal transport plans. This works best in applications that require solving multiple OT problems with shared structure. We instantiated it to speed up entropic regularized optimal transport and unregularized optimal transport with squared cost by multiple orders of magnitude. We envision extensions of the work in:

1. **Meta OT models**. While we mostly consider models based on hypernetworks, other meta-learning paradigms can be connected in. In the discrete setting, we only considered settings where the cost remains fixed, but the Meta OT model can also be conditioned on the cost by considering the entire cost matrix as an input (which may be too large for most models to handle), or considering a lower-dimensional parameterization of the cost that changes between the Meta OT problem instances.

2. **OT algorithms**. While we instantiated models on top of log-Sinkhorn and W2GN, Meta OT could be built on top of other methods.

3. **OT applications** that are computationally expensive and repeatedly solved, e.g. in multi-marginal and barycentric settings, or for Gromov-Wasserstein distances between metric-measure spaces.

**Limitations.** While we have illustrated successful applications of Meta OT, it is also important to understand the limitations: 1) **Meta OT does not make previously intractable problems tractable.** All of the baseline OT solvers we consider solve our problems within milliseconds or seconds. 2) **Out-of-distribution generalization.** Meta OT may not generate good predictions on instances that are not close to the training OT problems from the meta-distribution $\mathcal{D}$ over the measures and cost. If the model makes a bad prediction, one fallback option is to re-solve the instance from scratch.

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

# A  Selecting $\epsilon$ for MNIST

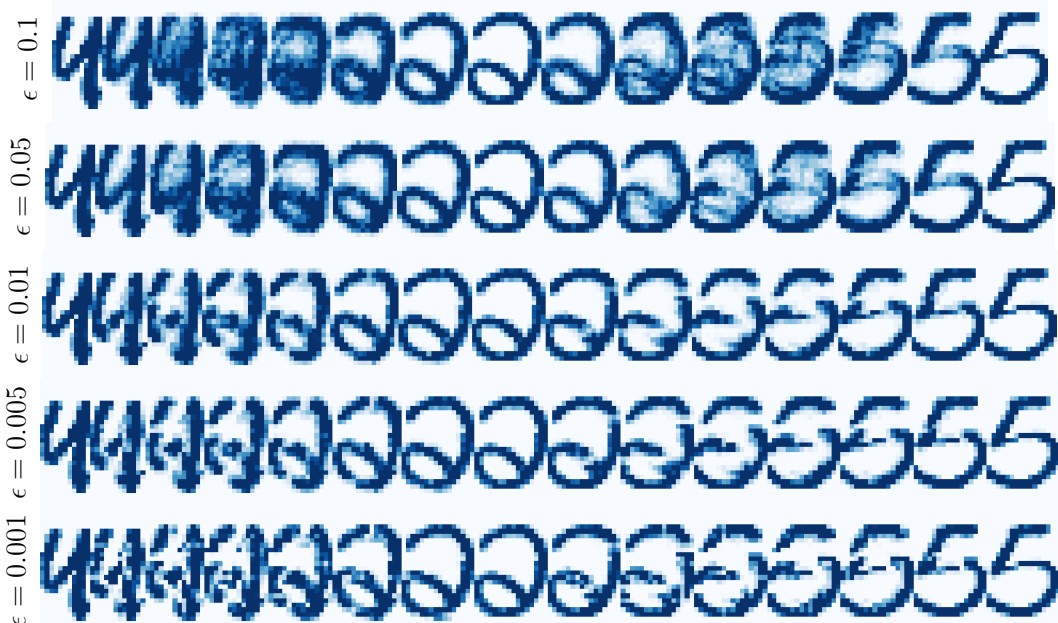

Figure 8: We selected $\epsilon = 10^{-2}$ for our MNIST coupling experiments as it results in transport maps that are not too blurry or sharp.

# B  Other models for continuous OT

While developing the hyper-network or Meta ICNN in sect. 3.2 for predicting couplings between continuous measures, we considered alternative modeling formulations briefly documented in this section. We finalized only the hyper-network model because it is conceptually the most similar to predicting the optimal dual variables in the continuous setting and results in rapid predictions.

## B.1  Optimization-based meta-learning (MAML-inspired)

The model-agnostic meta-learning setup proposed in MAML [Finn et al., 2017] could also be applied in the Meta OT setting to learn an adaptable initial parameterization. In the continuous setting, one initial version would take a parameterized dual potential model $\psi_\varphi(x)$ and seek to learn an initial parameterization $\varphi_0$ so that optimizing a loss such as the W2GN loss $\mathcal{L}$ from eq. (12) results in a minimal $\mathcal{L}(\varphi_K)$ after adapting the model for $K$ steps. Formally, this would optimize:

$$\arg\min_{\varphi_0} \mathcal{L}(\varphi_K) \quad \text{where} \quad \varphi_{t+1} = \varphi_t - \nabla_\varphi \mathcal{L}(\varphi_t) \tag{19}$$

Tancik et al. [2021] explores similar learned initializations for coordinate-based neural implicit representations for 2D images, CT scan reconstruction, and 3d shape and scene recovery from 2D observations.

**Challenges for Meta OT.** The transport maps given by $T = \nabla \psi$ can significantly vary depending on the input measures $\alpha, \beta$. We found it difficult to learn an initialization that can be rapidly adapted, and optimizing eq. (19) is more computationally expensive than eq. (18) as it requires unrolling through many evaluations of the transport loss $\mathcal{L}$. And, we found that *only* learning to predict the optimal parameters with eq. (18), conditional on the input measures, and then fine-tuning with W2GN to be stable.

**Advantages for Meta OT.** Exploring MAML-inspired methods could further incorporate the knowledge that the model's prediction is going to be fine-tuned into the learning process. One promising

direction we did not try could be to integrate some of the ideas from LEO [Rusu et al., 2018] and CAVIA [Zintgraf et al., 2019], which propose to learn a latent space for the parameters where the initialization is also conditional on the input.

## B.2 Neural process and conditional Monge maps

The (conditional) neural process models considered in Garnelo et al. [2018b,a] can also be adapted for the Meta OT setting, and is similar to the model proposed in Bunne et al. [2022a]. In the continuous setting, this would result in a dual potential that is also conditioned on a representation of the input measures, e.g. $\psi_\varphi(x; z)$ where $z := f_\varphi^{\mathrm{emb}}(\alpha, \beta)$ is a learned embedding of the input measures that is learned with the parameters of $\psi$. This could be formulated as

$$\arg\min_\varphi \mathbb{E}_{(\alpha,\beta)\sim\mathcal{D}} \mathcal{L}(\varphi, f_\varphi^{\mathrm{emb}}(\alpha, \beta)), \tag{20}$$

where $\mathcal{L}$ modifies the model used in the loss eq. (12) to also be conditioned on the context extracted from the measures.

**Challenges for Meta OT.** This raises the issue on best-formulating the model to be conditional on the context. One way could be to append $z$ to the input point $x$ in the domain. Bunne et al. [2022a] proposes to use the Partially Input-Convex Neural Network (PICNN) from [Amos et al., 2017] to make the model convex with respect to $x$ and not $z$.

**Advantages for Meta OT.** A large advantage is that the representation $z$ of the measures $\alpha, \beta$ would be significantly lower-dimensional than the parameters $\varphi$ that our Meta OT models are predicting.

## C  Additional experimental and implementation details

We have attached the Jax source code necessary to run and reproduce all of the experiments in our paper and will open-source all of it. Here is a basic overview of the files:

```
meta_ot    Meta OT Python library code
    conjugate.py    Exact conjugate solver for the continuous setting
    data.py
    models.py
    utils.py
config    Hydra configuration for the experiments (containing hyper-parameters)
train_discrete.py    Train Meta OT models for discrete OT
train_color_single.py    Train a single ICNN with W2GN between 2 images (for debugging)
train_color_meta.py    Train a Meta ICNN with W2GN
plot_mnist.py    Visualize the MNIST couplings
plot_world_pair.py    Visualize the spherical couplings
eval_color.py    Evaluate the Meta ICNN in the continuous setting
eval_discrete.py    Evaluate the Meta ICNN for the discrete tasks
```

Connecting to the data is one difficulty in running the experiments. The easiest experiment to re-run is the MNIST one, which will automatically download the dataset:

```
./train_discrete.py # Train the model, outputting to <exp_dir>
./eval_discrete.py <exp_dir> # Evaluate the learned models
./plot_mnist.py <exp_dir> # Produce further visualizations
```

## C.1 Hyper-parameters

We briefly summarize the hyper-parameters we used for training, which we did not extensively tune. In the discrete setting, we use the same hyper-parameters for the MNIST and spherical settings.

Table 3: Discrete OT hyper-parameters.

| Name | Value |
|---|---|
| Batch size | 128 |
| Number of training iterations | 50000 |
| MLP Hidden Sizes | [1024, 1024, 1024] |
| Adam learning rate | 1e-3 |

Table 4: Continuous OT hyper-parameters.

| Name | Value |
|---|---|
| Meta batch size (for $\alpha, \beta$) | 8 |
| Inner batch size (to estimate $\mathcal{L}$) | 1024 |
| Cycle loss weight ($\gamma$) | 3. |
| Adam learning rate | 1e-3 |
| $\ell_2$ weight penalty | 1e-6 |
| Max grad norm (for clipping) | 1. |
| Number of training iterations | 200000 |
| Meta ICNN Encoder | ResNet18 |
| Encoder output size (both measures) | 256×2 |
| Meta ICNN Decoder Hidden Sizes | [512] |

## C.2 Sinkhorn convergence times, varying thresholds

In the main paper, table 1 reports the runtime of Sinkhorn to reach a convergence threshold of the marginal error being below a tolerance of $10^{-3}$, which is the default value used in many solvers. app. C.2 report the results from sweeping over other thresholds and show that Meta OT's initialization is consistently able to help.

Table 5: Sinkhorn runtime to reach a thresholded marginal error on MNIST.

| Initialization | Threshold=$10^{-2}$ | Threshold=$10^{-3}$ | Threshold=$10^{-4}$ | Threshold=$10^{-5}$ |
|---|---|---|---|---|
| Zeros | $4.5 \cdot 10^{-3}$ ±$1.5 \cdot 10^{-3}$ | $7.7 \cdot 10^{-3}$ ±$1.2 \cdot 10^{-3}$ | $1.1 \cdot 10^{-2}$ ±$1.8 \cdot 10^{-3}$ | $1.5 \cdot 10^{-2}$ ±$2.3 \cdot 10^{-3}$ |
| Gaussian | $4.1 \cdot 10^{-3}$ ±$1.2 \cdot 10^{-3}$ | $7.7 \cdot 10^{-3}$ ±$1.4 \cdot 10^{-3}$ | $1.1 \cdot 10^{-2}$ ±$1.7 \cdot 10^{-3}$ | $1.4 \cdot 10^{-2}$ ±$2.4 \cdot 10^{-3}$ |
| Meta OT | $2.3 \cdot 10^{-3}$ ±$9.2 \cdot 10^{-6}$ | $3.9 \cdot 10^{-3}$ ±$1.6 \cdot 10^{-3}$ | $6.7 \cdot 10^{-3}$ ±$1.4 \cdot 10^{-3}$ | $1.0 \cdot 10^{-2}$ ±$2.4 \cdot 10^{-3}$ |

Table 6: Sinkhorn runtime to reach a thresholded marginal error on the spherical transport problem.

| Initialization | Threshold=$10^{-2}$ | Threshold=$10^{-3}$ | Threshold=$10^{-4}$ | Threshold=$10^{-5}$ |
|---|---|---|---|---|
| Zeros | $8.8 \cdot 10^{-1}$ ±$1.3 \cdot 10^{-1}$ | $1.4$ ±$1.9 \cdot 10^{-1}$ | $2.1$ ±$3.6 \cdot 10^{-1}$ | $2.8$ ±$5.6 \cdot 10^{-1}$ |
| Gaussian | $5.6 \cdot 10^{-1}$ ±$9.9 \cdot 10^{-2}$ | $1.1$ ±$2.0 \cdot 10^{-1}$ | $1.7$ ±$3.5 \cdot 10^{-1}$ | $2.4$ ±$5.4 \cdot 10^{-1}$ |
| Meta OT | $7.8 \cdot 10^{-2}$ ±$3.4 \cdot 10^{-2}$ | $0.44$ ±$1.5 \cdot 10^{-1}$ | $0.97$ ±$3.2 \cdot 10^{-1}$ | $1.7$ ±$6.8 \cdot 10^{-1}$ |

 **C.3   Experimental runtimes and convergence**

App. C.3 shows the convergence during training of Meta OT models in the discrete and continuous
settings over 10 trials on our single Quadro GP100 GPU. The MNIST models are consistently trained
to optimality within 2 minutes (!) while the continuous model takes a few hours to train.

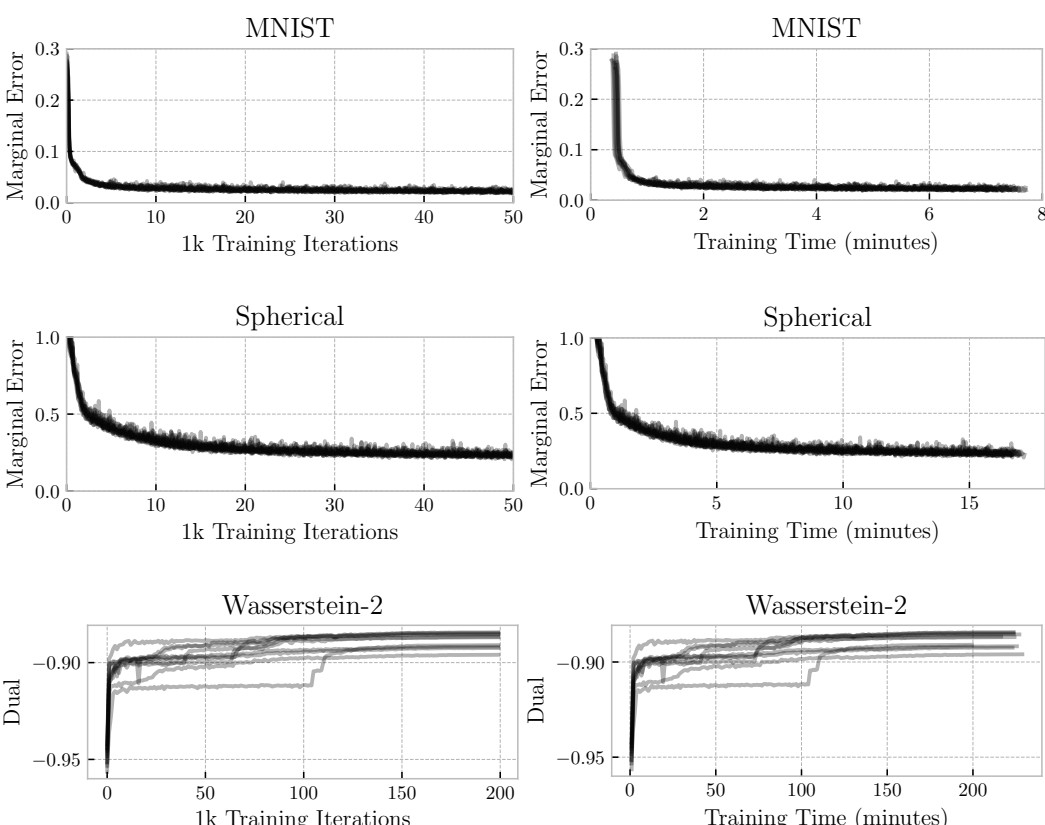

Figure 9: Convergence of Meta OT models during training, reported over iterations and wall-clock
time. We run each experiment for 10 trials with different seeds and report each trial as a line.

# D  Out-of-distribution generalization

App. D tests the ability of Meta OT to predict potentials for out-of-distribution input data. We consider the pairwise training and evaluation on the following datasets: 1) MNIST; 2) USPS [Hull, 1994] (upscaled to have the same size as the MNIST); 3) Google Doodles dataset* with classes Crab, Cat and Faces; 4) sparsified random uniform data in [0,1] where sparsity (zeroing values below 0.95) is used to mimic the sparse signal in black-and-white images. For each pair, eg, MNIST-USPS, we train on one dataset and use the other to predict the potentials. The comparison is done using the same metric as before, i.e., the deviation from the marginal constraints defined in eq. (7).

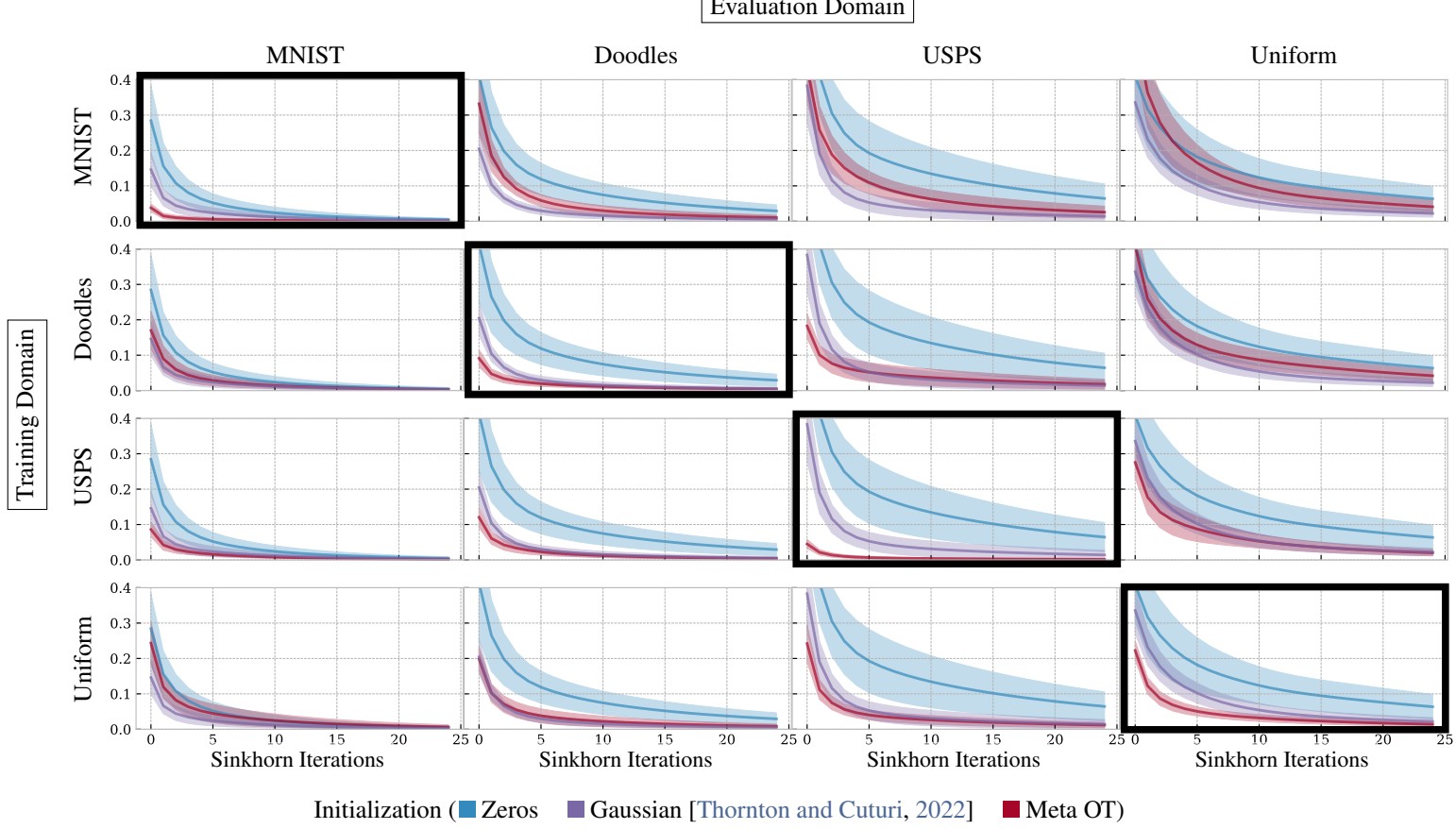

Figure 10: Cross-domain experiments.

---

*https://quickdraw.withgoogle.com/data

## E  Additional color transfer results

We next show additional color transfer results from the experiments in sect. 4.3 on the following public domain images from WikiArt:

- Distant View of the Pyramids by Winston Churchill (1921)
- Charing Cross Bridge, Overcast Weather by Claude Monet (1900)
- Houses of Parliament by Claude Monet (1904)
- October Sundown, Newport by Childe Hassam (1901)
- Landscape with House at Ceret by Juan Gris (1913)
- Irises in Monet's Garden by Claude Monet (1900)
- Crystal Gradation by Paul Klee (1921)
- Senecio by Paul Klee (1922)
- Váza s květinami by Josef Capek (1914)
- Sower with Setting Sun by Vincent van Gogh (1888)
- Three Trees in Grey Weather by Claude Monet (1891)
- Vase with Daisies and Anemones by Vincent van Gogh (1887)

$\alpha$  $\beta$  $T_{\#}\alpha$  $T_{\#}^{-1}\beta$

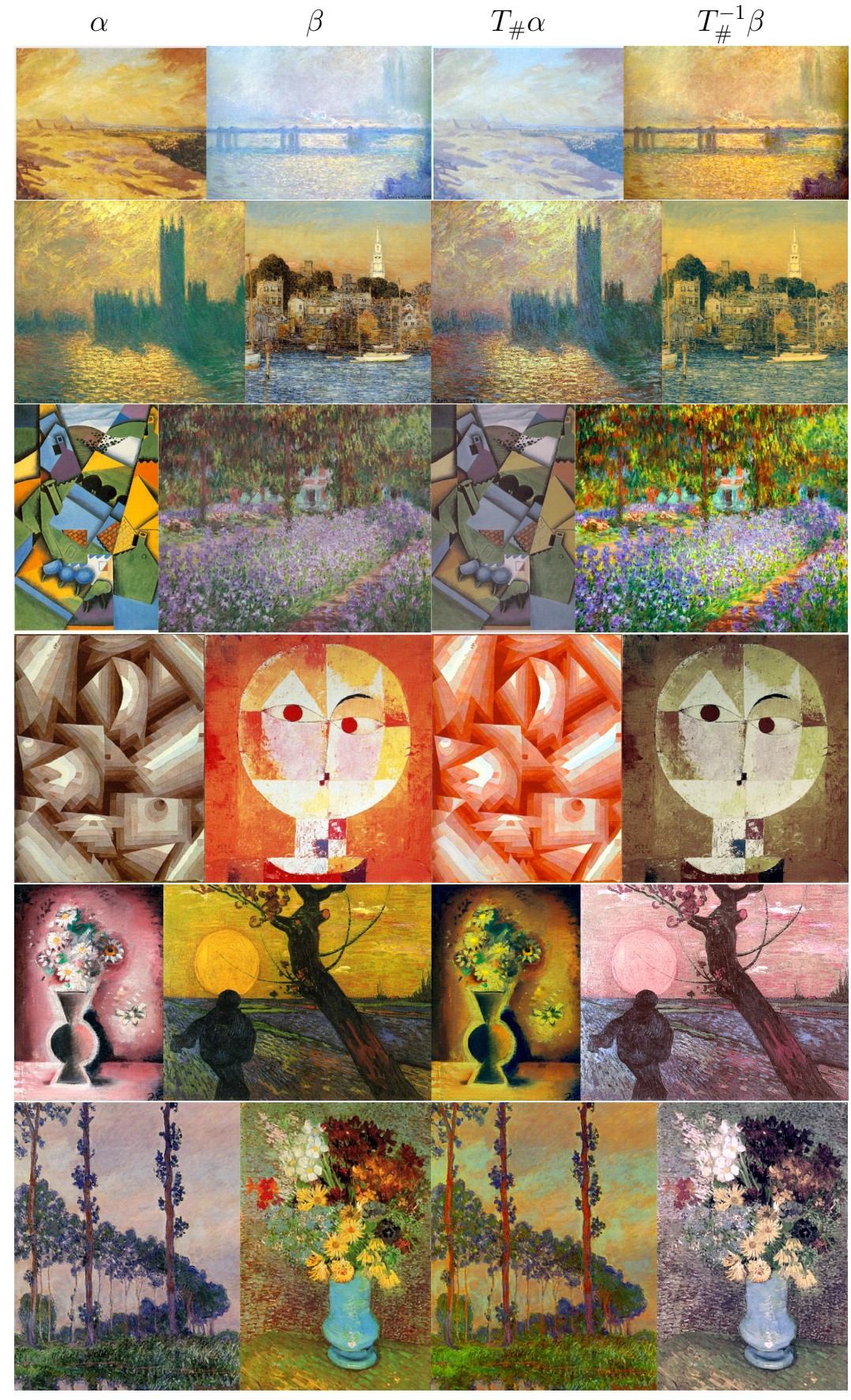

Figure 11: Meta ICNN (initial prediction). The sources are given in the beginning of app. E.

$\alpha$ $\qquad\qquad$ $\beta$ $\qquad\qquad$ $T_{\#}\alpha$ $\qquad\qquad$ $T_{\#}^{-1}\beta$

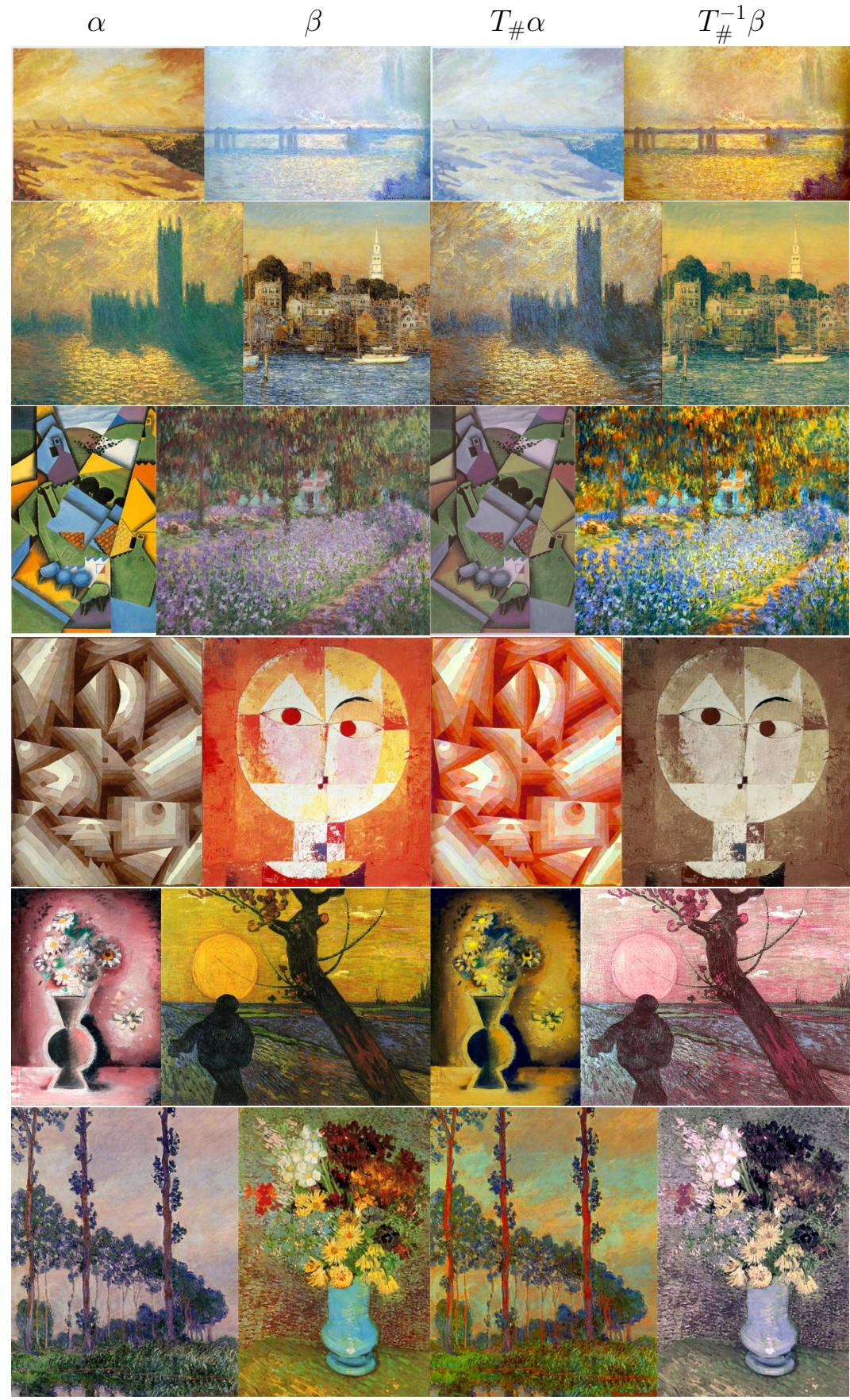

Figure 12: Meta ICNN + W2GN fine-tuning. The sources are given in the beginning of app. E.

$\alpha$ $\beta$ $T_{\#}\alpha$ $T_{\#}^{-1}\beta$

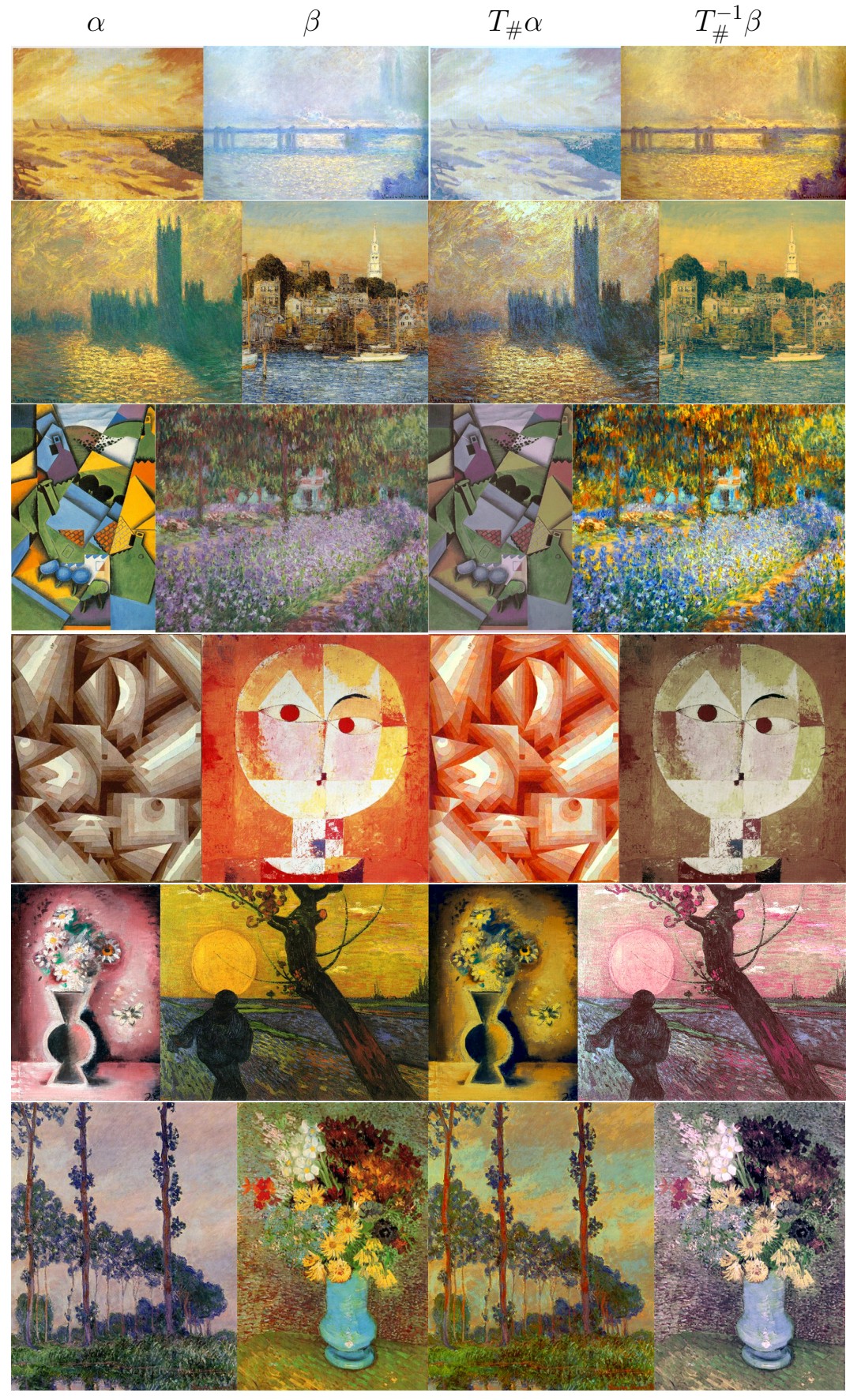

Figure 13: W2GN (final). The sources are given in the beginning of app. E.

