# OpenReview forum: "Meta Optimal Transport"
_NeurIPS.cc/2022/Conference — NeurIPS 2022 Submitted_

### Official Review · Reviewer_Yc4m · 2022-06-20

**Rating:** 7
**Confidence:** 4
**Soundness:** 3 good
**Presentation:** 3 good
**Contribution:** 2 fair

**Summary:**

This paper views the dual problem of optimal transport via the lens of amortized optimization. Then the authors propose a novel method to efficiently predict the optimal transport maps from the input measures. The predicted solutions can be then used as an initialization for standard OT solvers. Empirically, this method improves the computational time of standard OT solvers by multiple orders of magnitude in both discrete and continuous settings.


**Questions:**

I have the following questions:
1. The sentence in L158 is not clear. Could you explain it in more detail?
2. What is the formula for the marginal error in Table 1 and Figure 4? Is it Equation (7)? What if we use a stricter threshold like 10e-4, 10e-5, etc. Does MetaOT + Sinkhorn still converge faster?
3. What is the formula for the normalized dual objective value in Figure 7? Is 1.0 the optimized value?
4. **Related work.** I came across a recently published paper “Rethinking Initialization of the Sinkhorn Algorithm”. Because both papers try to generate an efficient initialization for speeding up the Sinkhorn algorithm, it may be useful to cite and discuss that paper.

**Minors:**
* *Sinkhorn algorithm.* The update for $g_i$ in Algorithm 1 should use $f_i$ instead of $f_{i-1}$.


**Limitations:**

The authors provided two limitations of Meta OT which may be practically important but are out of their problem settings. Other than those two, it would be interesting to see the applications of Meta OT on deep learning tasks such as generative modeling or domain adaptation.


**Strengths And Weaknesses:**

While the use of amortization in OT is not completely new, this work proposed a novel approach to find good initializations to solve multiple OT problems more efficiently. Experimental results back the computational advantages of Meta OT. Using the initial prediction from Meta OT, standard OT solvers converge much faster. However, there is no demonstration of some standard machine learning tasks that famously gain benefits from OT (e.g. generative modeling, domain adaptation). Therefore, it is difficult to judge the “quality” of transport maps of the proposed solvers in practical applications. Overall, this paper is well-structured and easy to follow. There are some minor parts that can be improved to increase the readability of the paper.

---

> ### Author Response · Authors · 2022-08-02
> **Author response part 1/2**
>
> Many thanks for the careful reading of our paper! We have attached a new version of the paper that we hope will clarify the points that you have raised, and include a more detailed response inline below. Please let us know if there are any other ablations or updates you think would be useful to get in.
>
> We would like to start by addressing a **misunderstanding** in the applicability of Meta OT methods and emphasize that **Meta OT methods may not be useful in every OT setting**. Your review requests that we evaluate in the standard image and generative modeling settings and state that not including these results is a weakness and limitation of our paper. These settings are out-of-scope and we do not see a useful way of incorporating Meta OT into these settings as they often seek to estimate a single transport map, or they only care about estimating the transport map between the latest model and the data distribution (and make use of warm-starting the dual potentials). We have added a new section to the introduction entitled “settings that are not Meta OT” to attempt to prevent this misunderstanding in the future and will respond in more detail to your points inline below. We hope that you will reconsider your evaluation of our paper in light of this clarification.
>
> Here are further responses inline:
>
> > While the use of amortization in OT is not completely new
>
> We are not aware of previous work using amortization for computing OT duals. We have a brief discussion of [Amortized Projection Optimization](https://arxiv.org/abs/2203.13417): they use amortized optimization to compute informative directions to project on in the context of *sliced* Wasserstein distances. The use case is deeply different from ours predicting the optimal duals given the input measures. Please let us know if you have any other references in mind.
>
> > However, there is no demonstration of some standard machine learning tasks that famously gain benefits from OT (e.g. generative modeling, domain adaptation). Therefore, it is difficult to judge the “quality” of transport maps of the proposed solvers in practical applications.
>
> Please see our earlier comment on these not being Meta OT settings and let us know if you have any other questions or comments on this. We hope that you will reconsider your evaluation in light of this.
>
> > There are some minor parts that can be improved to increase the readability of the paper.
>
> We hope you find the new version of our paper to be improved. Please let us know if there are any other updates you would like for us to get in.
>
> > The sentence in L158 is not clear. Could you explain it in more detail?
>
> This is on the delicate topic of comparing the convergence of Meta OT to a classical method, such as Sinkhorn. We thought more about this part of the paper and decided to remove it for now, as Meta OT methods are about prediction rather than convergence. Our intention in this part was to say that amortized optimization provides a ~constant-time (and hopefully computationally cheap) way of predicting solutions to optimization problems that are otherwise iteratively solved. The standard theoretical convergence analysis results applied to, for example, gradient descent or Sinkhorn considers the rate the iterates approach the optimal solution: the point we would like to make here is that a Meta OT model’s prediction is often significantly computationally cheaper than running an iterative algorithm to the same level of accuracy. This is because the Meta OT model only amortized a subspace of OT problems. Please let us know if you have any outstanding questions or comments on this point, or if you would find it insightful for us to include anything back in the paper.
>
> > What is the formula for the marginal error in Table 1 and Figure 4? Is it Equation (7)? What if we use a stricter threshold like 10e-4, 10e-5, etc. Does MetaOT + Sinkhorn still converge faster?
>
> Yes, it’s the marginal error in eq. (7). We have updated this in the text. We arbitrarily selected 1e-3 as the convergence threshold because it is a commonly used default value. We have ablated other values of 1e-2, 1e-4, and 1e-5 in Table 5 and 6 in Appendix C.2 and show that Meta OT’s initialization improves the runtime in all cases.
>
> > What is the formula for the normalized dual objective value in Figure 7? Is 1.0 the optimized value?
>
> We estimate the dual objective by exactly conjugating the model and then normalize the value for each instance by the smallest and largest values encountered during the W2GN fine-tuning so that the instances are comparable. Without the normalization, the optimal dual objective between the color palette transfers can be significantly different and make it difficult to easily compare how the methods converge.

---

> > ### Author Response · Authors · 2022-08-02
> > **Author response part 2/2**
> >
> > > Related work. I came across a recently published paper “Rethinking Initialization of the Sinkhorn Algorithm”. Because both papers try to generate an efficient initialization for speeding up the Sinkhorn algorithm, it may be useful to cite and discuss that paper.
> >
> > This is a brilliant paper! It was posted after the submission deadline and proposes an important idea to compare to: using the Wasserstein-2 OT between Gaussian approximations of the measures to initialize the Sinkhorn dual potentials rather than predicting an initialization with Meta OT. We have added a comparison to their approach in all of the relevant settings (in L205-209 Figure 4 and Figure 7) and find that Meta OT predictions often provide better starting points. We also note that their initial setting is mostly scoped to the Euclidean Wasserstein-2 setting while Meta OT methods do not make this assumption: this makes their method not exactly applicable in our spherical setting, which uses the spherical geodesic cost rather than the Euclidean one. We find that their initialization is still useful in the spherical setting as the Euclidean distance in the ambient space is correlated with the geodesic distance. (Very tangentially, one other interesting idea for their setting here is that a Riemannian extension of their method could be created too, that looks at known Riemannian OT-based initializations between, for example, wrapped Gaussians on the sphere.)
> >
> > > Other than those two, it would be interesting to see the applications of Meta OT on deep learning tasks such as generative modeling or domain adaptation.
> >
> > Please see our earlier comment on these not being Meta OT settings and let us know if you have any other questions or comments on this. We hope that you will reconsider your evaluation in light of this.

---

> > > ### Comment · Reviewer_Yc4m · 2022-08-03
> > > **Response to Authors**
> > >
> > > I thank the authors for their response.
> > >
> > > After reading their rebuttal, I believe that the authors have adequately addressed all of my questions. In addition, the revised version of the paper does improve both the quality and clarity of the paper. My only concern is still about the practical applications of Meta OT.
> > >
> > > All in all, I would like to increase my score from 6 to 7.

---

### Official Review · Reviewer_96EJ · 2022-07-10

**Rating:** 6
**Confidence:** 4
**Soundness:** 3 good
**Presentation:** 4 excellent
**Contribution:** 3 good

**Summary:**

As an experimental paper, Meta OT provides better initialization to OT solvers by leveraging shared representations from previous tasks. It helps accelerate the training of OT solvers which is well supported by experimental results provided in this paper. One key requirement is that downstream tasks must follow the identical distribution as the previous tasks. This was acknowledged by the authors under the limitations of meta OT.

In general, the paper is concise and well-written. The main idea is clear. My major concerns are listed in the main review. I am willing to raise my score if these concerns are properly addressed.



**Questions:**

Please see **Main Review** and **Weaknesses** for questions to address during the rebuttal.

**Limitations:**

### Weaknesses:
1. In checklist 2, the authors mention that the paper is not a theory paper. In my opinion, the paper currently lacks sufficient experimentation that may vouch for its clear acceptance. I suggest the authors include stronger experimental results on CIFAR10, CelebA, CelebA-HQ, or other relevant datasets. If time permits, the authors may choose the task of generative modeling or image restoration where OT has shown promising results. How efficiently can Meta OT predict the parameters of the OT map in the aforementioned tasks on these harder datasets?
2. The paper claims that we would receive a stream of new images in deployment which could be different from the images used to obtain the OT map (lines 31-32). However, the individual experiments on MNIST, Spherical, and WikiArt are conducted with samples from the same distribution. I understand that while standard OT solvers need retraining from scratch, Meta OT provides a better initialization. The authors should discuss to what extent Meta OT can handle the stream of new images in deployment.
3. It would be helpful to assess the performance of color transfer if the authors provide quantitative results. It is hard to judge qualitatively since images visually look all the same. I suggest both perception and distortion metrics to analyze perceptual quality and geometric distortion in the pushforward samples.

**Strengths And Weaknesses:**

## Main Review
### Strengths
1. The idea of predicting the parameters of convex Brenier potentials approximated by ICNNs is a major strength due to the following reasons. First, directly learning the parameters of an ICNN while satisfying the convexity constraint is shown to be challenging in prior works. Second, the gradient of the ICNN may not contain the true OT map due to its poor expressive power. This paper takes a step towards resolving the first challenge by learning to predict its parameters, e.g. through a ResNet and MLP. As an extension, one might increase the complexity of ICNNs which may resolve the second challenge.
2. While prior works have tried to directly approximate the gradient of Brenier potentials using a neural network, there is insufficient theoretical evidence on whether the learned neural network is indeed an OT map. On the other hand, this paper uses a meta-learning approach to directly learn the parameters of Brenier potentials. Thereby, it ensures the optimality of the learned transport map as per Brenier’s theorem.

### Paper organization/presentation
1. The paper is well organized and nicely written.
2. The presentation is clear and concise.

### Experiments
1. At present, the experiments are conducted on relatively easier datasets.
2. The lack of quantitative results makes it hard to evaluate the overall performance.
3. Ablation study is needed to measure the distance of fully trained weights from random initialization and Meta OT predicted initialization. Whether a better initialization would give results comparable to Meta OT?

### References
1. The paper includes a comprehensive list of relevant literature.
2. One suggestion would be to cite the published version of papers where applicable.

---

> ### Author Response · Authors · 2022-08-02
> **Author response part 1/2**
>
> Many thanks for the careful reading of our paper! We have attached a new version of the paper that we hope will clarify the points that you have raised, and include a more detailed response inline below. Please let us know if there are any other ablations or updates you think would be useful to get in.
>
> We would like to start by addressing a **misunderstanding** in the applicability of Meta OT methods and emphasize that **Meta OT methods may not be useful in every OT setting**. Your review requests that we evaluate in the standard image and generative modeling settings and state that not including these results is a weakness and limitation of our paper. These settings are out-of-scope and we do not see a useful way of incorporating Meta OT into these settings as they often seek to estimate a single transport map, or they only care about estimating the transport map between the latest model and the data distribution (and make use of warm-starting the dual potentials). We have added a new section to the introduction entitled “settings that are not Meta OT” to attempt to prevent this misunderstanding in the future and will respond in more detail to your points inline below. We hope that you will reconsider your evaluation of our paper in light of this clarification.
>
> Here are further responses inline:
> > The lack of quantitative results makes it hard to evaluate the overall performance.
>
> Many of our results are quantitative: Table 1 quantifies the runtime of Meta OT in comparison to standard Sinkhorn solves (with zero and Gaussian init) when using the default marginal error of 1e-3, Table 2 quantifies the runtime and dual objective values in comparison to standard W2GN solves, Figure 4 quantitatively shows the marginal error of Meta OT on test data in comparison to Sinkhorn (with zero and Gaussian init). Figure 7 shows the dual objective values in comparison to W2GN on test data. And our new version of the paper contains a few more quantitative details: Table 5 and 6 in the appendix quantitatively compare the runtime against Sinkhorn when varying the convergence threshold, and Figure 10 in the appendix quantitatively compares cross-domain training/evaluations to asses the generalization capabilities of Meta OT methods.
>
> Are there any other specific quantitative settings and evaluations you were referring to in your original comment that you would be curious to see?
>
> > Ablation study is needed to measure the distance of fully trained weights from random initialization and Meta OT predicted initialization.
>
> We would be open to running additional experiments for ablations like this. If you would be interested in seeing these results, can you please clarify what you would expect to see in this ablation?
>
> > Whether a better initialization would give results comparable to Meta OT?
>
> When we submitted the paper, we were not aware of better initialization strategies. The paper [Rethinking Initialization of the Sinkhorn Algorithm](https://arxiv.org/abs/2206.07630) was posted after we submitted this, and we think it is a brilliant idea. We have added a comparison to the Gaussian initializations proposed here. We find that it is indeed able to improve upon Sinkhorn’s initialization and Meta OT’s predictions still further improve upon those. Please let us know if there are any other initialization schemes that you are aware of that would also make sense for us to include.
>
> > One suggestion would be to cite the published version of papers where applicable.
>
> Thanks, we have went through and updated the citations to the published versions. Please let us know if we have missed any of them.
>
> >  In my opinion, the paper currently lacks sufficient experimentation that may vouch for its clear acceptance. I suggest the authors include stronger experimental results on CIFAR10, CelebA, CelebA-HQ, or other relevant datasets. If time permits, the authors may choose the task of generative modeling or image restoration where OT has shown promising results. How efficiently can Meta OT predict the parameters of the OT map in the aforementioned tasks on these harder datasets?
>
> Please see our earlier comment on these not being Meta OT settings and let us know if you have any other questions or comments on this. We hope that you will reconsider your evaluation in light of this.

---

> > ### Author Response · Authors · 2022-08-02
> > **Author response part 2/2**
> >
> > > The paper claims that we would receive a stream of new images in deployment which could be different from the images used to obtain the OT map (lines 31-32). However, the individual experiments on MNIST, Spherical, and WikiArt are conducted with samples from the same distribution. I understand that while standard OT solvers need retraining from scratch, Meta OT provides a better initialization. The authors should discuss to what extent Meta OT can handle the stream of new images in deployment.
> >
> > While “new stream of images” can have many interpretations, in this informal phrasing we meant that a new stream of images that’s close to the i.i.d. samples from the meta-distribution used to train the model. It is indeed important in practice for any machine learning system to adapt to a stream of data that is not producing i.i.d. samples and is also likely to have distribution shift: we did not intend to claim any improvements in these settings and are very open to rephrasing any parts of our paper to make it clear that we are not addressing this.
> >
> > > It would be helpful to assess the performance of color transfer if the authors provide quantitative results. It is hard to judge qualitatively since images visually look all the same. I suggest both perception and distortion metrics to analyze perceptual quality and geometric distortion in the pushforward samples.
> >
> > We believe this is another minor misunderstanding, as Figure 7 quantitatively shows the dual objective on the color transfer between test images in comparison to the convergence of standard W2GN training, and Table 2 compares the runtime and dual values. We agree our contribution would be difficult to assess without this. We also hope that you are willing to re-evaluate your assessment of our paper in light of this new information.

---

> > > ### Comment · Reviewer_96EJ · 2022-08-09
> > > **Rebuttal Response**
> > >
> > > **Specific Quantitative Results**
> > >
> > > It would be interesting to see quantitative results using distortion metrics such as PNSR or SSIM where reference images are available. Also, the authors may provide quantitative results using a perceptual metric such as LPIPS to better capture the quality of the transported image samples.
> > >
> > > **Regarding initialization**
> > >
> > > I would like to see if a better initialization would give results comparable to Meta OT. Since the *Rethinking initialization* paper appeared after the submission of this work, I think it is not necessary to compare it with that work at this moment.
> > >
> > > **Quantitative metrics**
> > >
> > > By quantitative metrics, I don't mean the OT distance or the runtime because it poorly correlates with the perceptual image quality. I suggest the authors may consider perception/distortion metrics which are commonly used in practice.
> > >
> > > **New stream of images**
> > >
> > > I kindly ask the authors to rephrase the sentence.
> > >
> > > I thank the authors for the detailed response. Having read the rebuttal, I am raising my score from 5 to 6.

---

> > > > ### Author Response · Authors · 2022-08-09
> > > > **Response**
> > > >
> > > > Thank you for the response! We will update the phrasing on the stream of images and continue to think more about other initializations and quantitative comparisons to make.

---

> ### Author Response · Authors · 2022-08-09
> **Reminder for 96EJ**
>
> Dear Reviewer 96EJ, as the discussion period will close tomorrow, we would greatly appreciate if you'd take a look at our response to your review soon and let us know if you have any remaining questions. We look forward to addressing any remaining concerns before the end of the discussion period. If our response was satisfactory, we ask that you consider raising your score for our submission. Thank you for your time.

---

### Official Review · Reviewer_uWt4 · 2022-07-11

**Rating:** 8
**Confidence:** 4
**Soundness:** 3 good
**Presentation:** 4 excellent
**Contribution:** 3 good

**Summary:**

In this work, the authors introduce a new and very efficient procedure to solve optimal transport problems between both discrete and continuous distributions using amortized optimization. More precisely, in the setting where one aims at solving multiple OT problems between distributions sharing similar structures, the authors propose to learn, in an unsupervised manner, a predictor able to infer the optimal coupling instead of solving from scratch each OT problem considered. Such learned function can then be applied to quickly predict the optimal coupling between two test distributions (in the sense that they have not been used in the training stage) sharing the same structures as the distributions considered in the training stage. To do so, the authors propose to learn a function parametrized by neural networks mapping the input distributions and the underlying cost to the optimal potential. In order to learn such function, the authors  minimize the expectation on a distribution of problems (defined by two input measures and an underlying cost) of the dual objective of the OT problem under the same constraints. Note that if the family of functions generated by the network is sufficiently expressive, then solving such problem is the same as solving each single OT problem and therefore finding the true optimal potential on each of these problems. Note also that the proposed method is completely unsupervised as they do not require to solve each OT problem considered in the training stage. In the discrete setting, the authors consider the dual formulation of the entropic OT which is an unconstrained problem, therefore they end up with an unconstrained optimization problem studied and propose a simple gradient-descent procedure to solve it. In the continuous setting, the authors restrict themselves to the learning of the Wasserstein-2 distance, and take advantage of a reformulation of its dual involving convex potential to recover an unconstrained optimization problem too. Indeed in this case, the authors aim at predicting an ICNN by learning a function mapping the input measures to the parameters of the ICNN. The authors propose also to refine the prediction on test problems by applying few steps of either the Sinkhorn algorithm in the discrete setting or of the W2GN algorithm in the continuous one. Finally, they show on various experiments on real-world data that the proposed approach is able to recover almost the true potentials (and couplings) while being much faster.


**Questions:**

How do you take into account that the learned dual potential is also a function of the underlying cost in the discrete setting? As it seems that the MLP takes only into account the atoms of the measures.

How does this method compare to the supervised approach where we consider the exact same parametrization of the potential however we aim at minimizing the distance between the predicted potential and the true one?


How long is the training stage?
It would be nice to provide for each experiment, the plot of the training loss againts the CPU/GPU time or the number of operations. It would be nice also to see how versatile can be the model in the sense that if the network considered is sufficiently large, then it may be able to learn the optimal potential in various settings and not only for a specific distribution of problems.


**Limitations:**

Yes.

**Strengths And Weaknesses:**

I think that the paper is of very good quality, very clear and concise, and that the contributions presented here may be of real interests for practitioners using OT. More precisely, I think that learning a map that given a geometry is able to infer the optimal coupling for a family of problems is a very promising line of work which may be useful in various setting. For example one could think at some networks which during the forward pass, aim at some point to align data and therefore for each forward pass, one OT problem has to be solved. Another example could be that some studies require to compute multiple OT problems between similar point clouds which can be very time-consuming. The approach proposed here could significantly improves the computational time of this kind of tasks. I think that the authors should even motivate more their contributions by providing some contexts where the proposed learning procedure could be useful.

---

> ### Author Response · Authors · 2022-08-02
> **Author response**
>
> We were delighted to receive your review of our paper! Thanks for the encouraging comments and insightful questions. We have attached a new version of the paper that we hope will clarify the points that you have raised, and include a more detailed response inline below. Please let us know if there are any other ablations or updates you think would be useful to get in.
>
> > I think that the authors should even motivate more their contributions by providing some contexts where the proposed learning procedure could be useful.
>
> We agree and have added a few more settings in the introduction: to comparing seismic signals as in Engquist et al. and for single-cell perturbations as in multiple Bunne et al. papers. We also think the repeated couplings that arise in reinforcement learning settings may be useful for Meta OT methods.
>
> > How do you take into account that the learned dual potential is also a function of the underlying cost in the discrete setting? As it seems that the MLP takes only into account the atoms of the measures.
>
> We have not strongly considered this setting as our applications did not require it, but there are a few options worth considering if it comes up in the future. We have added the following text discussing this to the conclusion:
> *In the discrete setting, we only considered settings where
> the cost remains fixed, but the Meta OT model can also be conditioned
> on the cost by considering the entire cost matrix as an input
> (which may be too large for most models to handle), or considering
> a lower-dimensional parameterization of the cost that changes between
> the Meta OT problem instances.*
>
> > How does this method compare to the supervised approach where we consider the exact same parametrization of the potential however we aim at minimizing the distance between the predicted potential and the true one?
>
> We used regression-based amortization onto the ground-truth potentials in our early prototypes in the discrete/Sinkhorn setting, but did not want to assume access to high-accuracy ground-truth solutions and switched to the objective-based approach presented throughout out paper. In the settings we consider, we do not think there are any insightful ablations between the regression- and objective-based losses.
>
> We did not try regressing in the continuous Wasserstein-2 setting as approximate ground-truth solutions would have taken ~2 seconds to obtain per pair of images by running W2GN. In contrast, our Meta OT model is able to make a prediction in 1ms on an image pair and locally improves the meta parameters directly by using the gradient of the dual objective.
>
> The design choice of the loss may be important to consider for future settings, and we have updated the text in Section 2.2 to mention this choice.
>
> > How long is the training stage? It would be nice to provide for each experiment, the plot of the training loss against the CPU/GPU time or the number of operations.
>
> We’ve added plots to appendix C.3 showing that the MNIST experiments consistently converge after 2 minutes (!) of training on our GPU while the color transfer experiment takes a few hours. We unfortunately do not have a good training metric to report when training W2GN as the loss there gives the correct gradients of the dual objective, but unfortunately does not have a meaningful value. We instead try to estimate the dual objective by numerically conjugating the ICNN potential on a set of training instances. While in the main paper we are able to normalize the dual objectives by the maximum and minimum values encountered for each instance, it is not easy to add this into our training code and we have left the dual values unnormalized in this comparison.
>
> > It would be nice also to see how versatile can be the model in the sense that if the network considered is sufficiently large, then it may be able to learn the optimal potential in various settings and not only for a specific distribution of problems.
>
> We’ve added some cross-domain experiments to appendix D where we find that in many cases, the learned Meta OT potentials can generalize beyond the distribution it is trained on. Here, we considered discrete OT problems between MNIST, Google Doodles, USPS, and random data and train and evaluate on every pairwise combination of these datasets.
>
> Perhaps one dream would be to train a Meta OT model on random data (perhaps even between measures of varying sizes) that is then able to be immediately useful for most downstream problems encountered. We are unfortunately not optimistic that a model like this would be possible to create, and even if it was, it may need to be prohibitively large. For now, we recommend and prefer to only train and evaluate Meta OT models in similar domains.

---

> > ### Comment · Reviewer_uWt4 · 2022-08-08
> > **Response to authors**
> >
> > Dear authors,
> >
> > Thank you for the very detailed response. The new version of the paper has addressed my questions/comments and I'm deciding to leave my score unchanged.

---

### Official Review · Reviewer_1F35 · 2022-08-19

**Rating:** 3
**Confidence:** 4
**Soundness:** 2 fair
**Presentation:** 3 good
**Contribution:** 2 fair

**Summary:**

This paper considers learning a meta-model to predict the solution to the OT problem.

**Questions:**

I have a question about the line 49-50 out-of-sample setting. If Meta OT can approximate the potential of W2GN well, then it should be able to apply to any samples in the considered task, not only just training samples.

**Limitations:**

see weakness

**Strengths And Weaknesses:**

The idea is novel and the paper is well-written.

Compared to other OT methods that run from scratch, it can save the time of learning OT potential for a new task. However, the experiments in this paper are only 2 or 3 dimensions. And I don't see how it can scale to a high dimension in the current framework (figure 3). For example, if $\alpha,\beta$ are a dataset of images and each image is a sample point, then $z_1$ and $z_2$ would be of very high dimension, and I conjecture MLP will not work anymore. And if it's only this kind of low dimension, for example, do people really care about the difference between 1e-5 seconds and 1e-3 seconds? I have a question mark about its real application.

Also, the pipeline in Figure 3 is quite problem-oriented. Given a new format of distributions $\alpha,\beta$ that are not images, for example, single-cell RNA data or a set of point clouds, one needs to re-design or re-select the encoder of $\alpha,\beta$. ResNet may not be able to extract the correct information.

Regarding the continuous measure $W_2$ solver, I think W2GN is not a good choice, at least not the best. It primarily considers OT maps in latent space of pre-trained autoencoder, which limits practical application in high dimensional space, see the discussion in the second paragraph of [Rout et al. 2021](https://openreview.net/pdf?id=5JdLZg346Lw). The best continuous $W_2$ solver has been shown to be MM:R in [W2 benchmark](https://openreview.net/pdf?id=CI0T_3l-n1) paper. Why do authors choose to use outdated W2GN instead of state-of-art $W_2$ solver?

Since the computation speed is a highlighted advantage, the authors should also consider the comparison with the GeomLoss package (https://www.kernel-operations.io/geomloss) and “Fast geometric learning with symbolic matrices” Feydy et al., NeurIPS 2020. It's a GPU implementation and it supports arbitrary cost functions and scales up to millions of high-dimensional samples in seconds.

The authors can be more careful about the literature:

Line 235: Li et al. 2020 didn't use ICNN. Instead, I find these two papers use ICNN

Alvarez-Melis, David, Yair Schiff, and Youssef Mroueh. "Optimizing functionals on the space of probabilities with input convex neural networks." Transactions on Machine Learning Research 2022.

Fan, Jiaojiao, Amirhossein Taghvaei, and Yongxin Chen. "Scalable computations of Wasserstein barycenter via input convex neural networks." ICML 2021.

---

### Author Response · Authors · 2022-08-02
**Shared author response**

We are grateful to the reviewers for carefully going through our paper, and are delighted to hear from the reviewers that *“the paper is of very good quality, very clear and concise, and that the contributions presented here may be of real interests for practitioners using OT”* (uWt4), as well as *“the idea of predicting the parameters of convex Brenier potentials approximated by ICNNs is a major strength”* (96EJ).

We are fully committed to incorporating all of the reviewer feedback the final version of our paper and have attached a new version of the paper that clarifies a few points that came up. We have run all of the additional experiments and analyses suggested by the reviewers and added the new results to the paper: 1) the Gaussian setting from [Rethinking Initialization of the Sinkhorn Algorithm](https://arxiv.org/abs/2206.07630) as a baseline to Table 1, Figure 4, and Figure 10, 2) cross-domain experiments to appendix D, 3) more training details on the convergence and runtimes to Appendix C.3, and 4) further ablations of the Sinhkorn convergence times as the threshold is varied in Appendix C.2.

We would also like to address a **misunderstanding** in the applicability of Meta OT methods and emphasize that **Meta OT methods may not be useful in every OT setting**. The reviews request that we evaluate in the standard image and generative modeling settings and state that not including these results is a weakness and limitation of our paper. These settings are out-of-scope and we do not see a useful way of incorporating Meta OT into these settings as they often seek to estimate a single transport map, or they only care about estimating the transport map between the latest model and the data distribution (and make use of warm-starting the dual potentials). We have added a new section to the introduction entitled “settings that are not Meta OT” to attempt to prevent this misunderstanding in the future and will respond in more detail to your points inline below. We hope that the reviewers will reconsider their evaluations of our paper in light of this clarification.

We will respond with more specific comments individually in the review threads.

---

### Meta-Review · Area_Chair_jKUh · 2022-08-29

**Recommendation:** Reject
**Confidence:** Less certain

**Metareview:**

This paper proposes an amortized optimization approach for predicting optimal transport (OT) maps. Three reviewers found that the proposed method is interesting. However, some concerns raised by another reviewer on the improvement of the computational efficiency and the generality of the proposed method were raised:

1) The experiments on computational efficiency are of very small scale, and insufficient to justify the improvement. The authors may consider larger scale problems, where the run time should be significantly larger than other computational overheads, e.g. >10 seconds.

2) As shown in Tables 1 and 2, the improvement of the proposed method on computational efficiency is marginal.

3) The current experiments only use MLP for small scale and relatively easy low dimensional problem. To demonstrate the generality of the proposed method, the authors should consider other neural architectures for more complex data.

This paper can be significantly strengthened if these issue could be addressed.

**Award:**

No

---

### Decision · Program_Chairs · 2022-09-14

Reject